# Electron transport chain biogenesis activated by a JNK-insulin-Myc relay primes mitochondrial inheritance in *Drosophila*

**Zong-Heng Wang, Yi Liu, Vijender Chaitankar, Mehdi Pirooznia, Hong Xu***

National Heart, Lung, and Blood Institute, National Institutes of Health, Bethesda, United States

**Abstract** Oogenesis features an enormous increase in mitochondrial mass and mtDNA copy number, which are required to furnish mature eggs with an adequate supply of mitochondria and to curb the transmission of deleterious mtDNA variants. Quiescent in dividing germ cells, mtDNA replication initiates upon oocyte determination in the *Drosophila* ovary, which necessitates active mitochondrial respiration. However, the underlying mechanism for this dynamic regulation remains unclear. Here, we show that an feedforward insulin-Myc loop promotes mitochondrial respiration and biogenesis by boosting the expression of electron transport chain subunits and of factors essential for mtDNA replication and expression, and for the import of mitochondrial proteins. We further reveal that transient activation of JNK enhances the expression of the insulin receptor and initiates the insulin-Myc signaling loop. This signaling relay promotes mitochondrial biogenesis in the ovary, and thereby plays a role in limiting the transmission of deleterious mtDNA mutations. Our study demonstrates cellular mechanisms that couple mitochondrial biogenesis and inheritance with oocyte development.

DOI: https://doi.org/10.7554/eLife.49309.001

*For correspondence:
hong.xu@nih.gov

Competing interests: The authors declare that no competing interests exist.

## Introduction

Mitochondria host a number of biosynthetic pathways and produce most of the cell's ATP through oxidative phosphorylation, which is carried out by the electron transport chain (ETC) complexes located on the mitochondrial inner membrane. While the majority of mitochondrial proteins are encoded on the nuclear genome, synthesized in the cytoplasm, and imported into the mitochondria, a subset of core ETC components are encoded on the mitochondrial genome (mtDNA) and synthesized inside the mitochondrial matrix. Thus, mitochondria biogenesis and ETC activity in particular, rely on the coordinated expression of both nuclear- and mtDNA-encoded mitochondrial genes (*Falkenberg et al., 2007*). Mitochondria vary in number and activity to meet the different energy and metabolic demands of different tissues and developmental processes. Mitochondria are transmitted exclusively through the maternal lineage in most metazoans (*Wallace, 2008*), which demands a complex regulation of mitochondrial biogenesis and ETC activity during oogenesis. Animal oocytes are hundreds of times larger than their progenitors (*Picton et al., 1998*). During this tremendous oocyte growth, mitochondria undergo prodigious biogenesis and increase mtDNA copy number over a thousand folds (*Stewart et al., 2008*). The massive amount of mitochondria in the mature oocyte is necessary to power early embryonic development, as inadequate mitochondrial contents often lead to embryonic lethality (*May-Panloup et al., 2007*). However, the mechanism by which the germline couples mitochondrial biogenesis to oocyte development remains elusive.

While furnishing mature oocytes with sufficient number of mitochondria, oogenesis also limits the transmission of harmful mtDNA mutations. The mitochondrial genome is prone to accumulating mutations because of its close vicinity to the highly mutagenic free radicals present in the mitochondrial matrix and of a lack of effective repair mechanisms (*Pesole et al., 1999*). Yet, harmful mtDNA mutations are rare in populations (*Stewart and Larsson, 2014*), underscoring the presence of efficient mechanisms to limit their transmission through the female germline. We previously reported that mtDNA replication depends on active respiration in the *Drosophila* ovary (*Hill et al., 2014*). Healthy mitochondria with wild-type genomes propagate more vigorously than defective ones carrying harmful mutations, thereby curbing the transmission of deleterious mtDNA mutations to the next generation (*Zhang et al., 2019*). Therefore, an active ETC appears to be a stress test for the functionality of mtDNA, and is essential for mtDNA selective inheritance. Nonetheless, how the activity of the ETC is regulated during oogenesis is not well understood.

Insulin signaling (IIS), an evolutionary conserved pathway that controls cell growth and proliferation (*Oldham and Hafen, 2003*), has also been shown to regulate ETC biogenesis and ATP production in human skeletal muscles (*Stump et al., 2003*). In the *Drosophila* ovary, IIS promotes the growth of follicles from the early to the middle stages of oogenesis (*LaFever and Drummond-Barbosa, 2005*). IIS activity decreases before the nurse cells dump their content into the oocyte. This decrease relieves the inhibition of GSK3, thereby shutting down mitochondrial respiration (*Sieber et al., 2016*). However, oogenesis begins with germline stem cells (GSCs) that are thought not to rely on oxidative phosphorylation to ATP production (*Kai et al., 2005*). We predicted there had to be developmental cues to activate mitochondrial respiration in the late germarium stage when mtDNA replication commences. IIS represents a logical candidate to modulate this metabolic transition in early oogenesis. Nonetheless, it remains to be explored how IIS is dynamically regulated during oogenesis and whether it is indeed involved in the aforementioned metabolic transition. Furthermore, little is known regarding how IIS modulates ETC activity and mtDNA biogenesis in general.

In this study, we find that mitochondrial respiration is quiescent in GSCs and dividing cysts, but markedly upregulated in the late germarium, the same spatial-temporal pattern as mtDNA replication. We uncover a feedforward loop between IIS and Myc protein which orchestrates the transcriptional activation of respiration and mtDNA replication. Furthermore, transient JNK activity boosts insulin receptor (InR) transcription to enhance the IIS-Myc loop. Our work uncovers how developmental programs couple mitochondrial biogenesis with cell growth and mitochondrial inheritance.

## Results

### Coordinated transcription of both nuclear and mitochondrial genome controls etc biogenesis

Mitochondrial DNA replication is significantly increased in the post-mitotic germ cells in late germarium and relies on the mitochondrial inner membrane potential ($\psi_m$) and etc activity (*Hill et al., 2014*). We therefore hypothesized that mitochondrial respiration might be developmentally regulated in a spatio-temporal pattern similar to that of mtdna replication. To test this idea, we monitored $\psi_m$, which is an indicator of mitochondrial respiration, in the developing germ cells. We found that $\psi_m$, measured as the ratio of tmrm (an indicator of membrane potential) to mitotracker green (an indicator of mitochondrial mass) (*Zhang et al., 2019*), was markedly higher in region 2b than at earlier stages in the germarium (*Figure 1A*), indicating that respiration is activated in the 16-cell cysts, concomitantly with the onset of mtdna replication. Consistently, etc activity, indicated by a dual sdh (succinate dehydrogenase)/cox (cytochrome c oxidase) colorimetric assay (*Ross, 2011*), was much higher in region 2b than at earlier germarium stages and remained high until the stage-10 egg chamber (*Figure 1B* and *Figure 1—figure supplement 1A*). These results suggest that etc activity is upregulated in the late germarium stages.

We next asked whether the dynamic pattern of ETC activity in the germarium reflected the expression of ETC subunits. Except for complex II (SDH) components, which are encoded on nuclear genome only, all other ETC proteins are encoded by both nuclear and mitochondrial genomes. Thus, we performed fluorescence in situ hybridization (FISH) with fluorescently labeled DNA probes specific to mRNAs of either nuDNA- or mtDNA-encoded ETC subunits in ovaries. Both COXIV

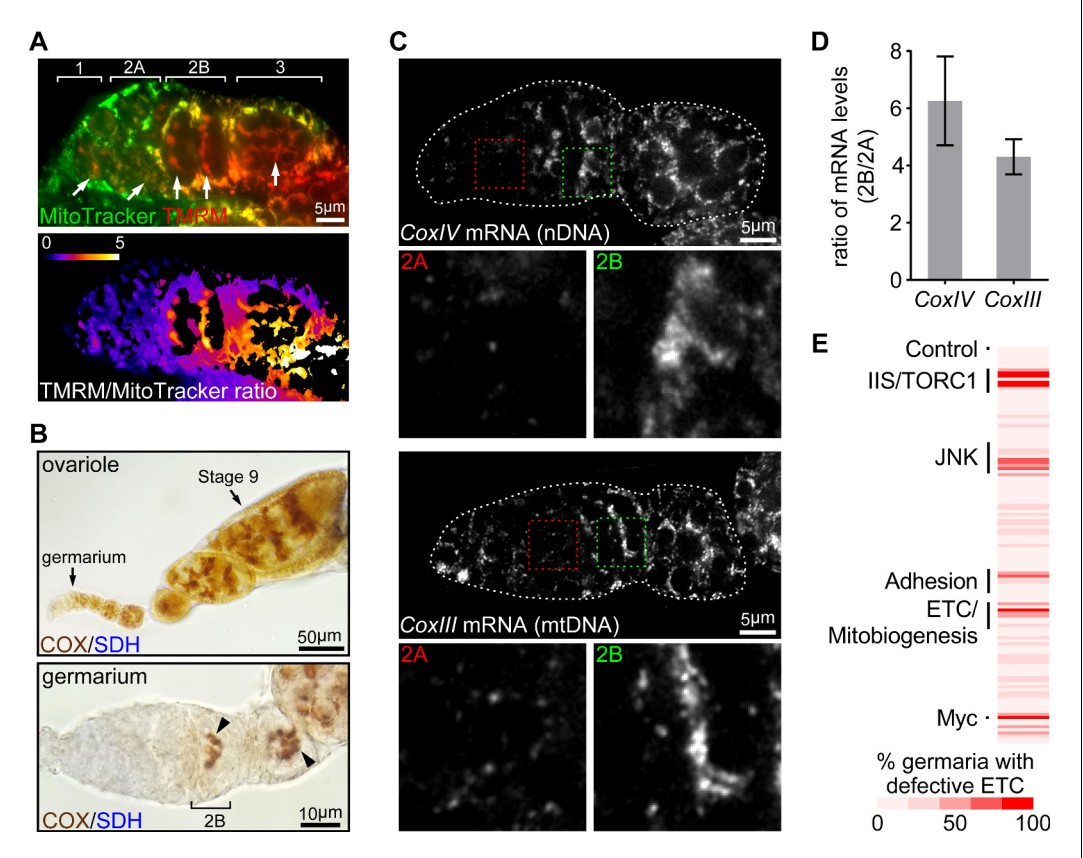

**Figure 1.** ETC activity and gene expression sharply increase at germarium stage 2B. (**A**) Upper panel: a representative image of a germarium stained with TMRM (a membrane potential marker) and MitoTracker Green (a mitochondrial mass marker). Germarium regions are indicated. Arrows indicate mitochondria in germ cells (GCs). Lower panel: TMRM/MitoTracker Green ratiometric image, indicating that mitochondria membrane potential in stem cells and dividing cysts is low, but markedly increased in 16-cell cysts and budding egg chambers. Scale bar, 5 μm. (**B**) Upper panel: a representative image of a wt ovariole (from germarium to stage nine egg chamber) stained for the COX/SDH dual activities. Lower panel: a representative high-magnification image of a germarium stained for COX/SDH. Note the onset of COX/SDH activity in region 2B of the germarium (arrowhead). Scale bars, 50 μm for the upper panel and 10 μm for the lower panel. (**C**) Visualization of the *CoxIV* and *CoxIII* mRNAs in germaria from wt flies by FISH with fluorescently labeled DNA probes. Germaria are outlined with dotted lines. For each mRNA labeling, lower panels illustrate the enlarged areas of germarium region 2A (red dotted line) and 2B (green dotted line), respectively, shown in the upper panels. Scale bars, 5 μm. (**D**) Quantification of the relative expression level of *CoxIV* or *CoxIII* mRNA in different regions of germarium. Note that both transcripts are markedly induced in region 2B germarium. Error bars represent SEM. (**E**) RNAi screen for genes required to induce ETC activity in germaria. For each RNAi line, the impact on ETC activity is scored as the percentage of germaria with reduced COX/SDH staining.

DOI: https://doi.org/10.7554/eLife.49309.002

The following source data and figure supplement are available for figure 1:

**Source data 1.** Ratio of mRNA in region 2B and region 2A of the germarium.
DOI: https://doi.org/10.7554/eLife.49309.004

**Figure supplement 1.** ETC activities at middle stages of oogenesis and expression of complex III genes in the germarium.
DOI: https://doi.org/10.7554/eLife.49309.003

(nuclear-encoded) and COXIII (mtDNA-encoded) transcripts exhibited low expression in earlier regions, but increased 4 to 6 folds in region 2B, recapitulating the pattern of ETC activity (*Figure 1C,D*). The same pattern was observed for Cyt-C1 (nuclear-encoded) and Cyt-B (mtDNA-encoded) (*Figure 1—figure supplement 1B*). These results indicate that the increased ETC activity detected with our COX/SDH colorimetric assay correlates with an increase of ETC genes expression at region 2B germarium. Taken together, these data suggest that the activation of respiration at stage 2B may partially relies on the coordinated transcription of nuDNA- and mtDNA-encoded genes.

## A candidate RNAi screen for upstream regulators of ETC biogenesis

To uncover the developmental cues that initiate ETC gene transcription in the late germarium, we screened a collection of 132 RNAi lines directed at major developmental pathways and at factors involved in cellular metabolism and mitochondrial functions (*Basson, 2012*; *Clavería and Torres, 2016*; *Desvergne et al., 2006*; *Perrimon et al., 2012*). We expressed dsRNAs to knockdown genes in the germ cells using a *nanos*-Gal4 (*nos*-Gla4) driver and applied the COX/SDH dual activity assay as an indirect measure of ETC abundance (*Ross, 2011*) (*Supplementary file 1*). We also included a few RNAi lines directed at COX components or genes essential for mitochondrial biogenesis as positive controls. As expected, knocking down these genes consistently impaired ETC activity (*Figure 1E*). Overall, 6 RNAi lines from the list caused germline degeneration and 12 lines led to reduced ETC activity without causing the loss of the germline or other defects in development. Among these 12 lines are components of the IIS/TORC1 signaling, the JNK pathway, cell adhesion molecules, translation regulators and one transcription factor (*Figure 1—figure supplement 1C*). Notably, all hits impaired activities of both COX and SDH, except for *coxV* RNAi, which disrupted COX activity only, indicating that the recovered genes are required for the expression of both nuclear and mitochondrial genes.

## Myc controls ETC biogenesis and mtDNA replication

The transcription factor Myc emerged as one of the strongest hits from our screen. Myc has been demonstrated to boost mitochondrial biogenesis and regulate energy metabolism in mammals (*Ahuja et al., 2010*; *Dang, 2013*; *Jellusova et al., 2017*; *Li et al., 2005*). Additionally, we found that Myc's expression pattern, monitored with a Myc-GFP fusion protein (*Greer et al., 2013*), mirrored the pattern of ETC activity in the ovary: low in the early stages, but elevated in germarium region 2B and remaining high until mid-stage egg chambers (*Figure 2A* and *Figure 2—figure supplement 1A*). These observations spurred us to explore the potential roles of Myc in the induction of ETC activity in region 2B cysts. To confirm the result of the Myc RNAi from the screen, we utilized a hypomorphic Myc allele, $myc^{P0}$, which has reduced level of *myc* mRNA, but progresses through early oogenesis (*Johnston et al., 1999*; *Quinn et al., 2004*). To get a semi-quantitative measure of COX activity, we visualized COX activity in the ovary (*Figure 2—figure supplement 2A,B*). Then, we generated a standard enzymatic activity curve for COX (*Figure 2—figure supplement 2C,D*) adopting previously established procedures (*Jung et al., 2002*; *Melendez-Ferro et al., 2013*). The activity of COX in the germarium was normalized to the COX activity on the standard curve. Consistent with the RNAi result, COX activity was markedly reduced in the $myc^{P0}$ ovaries, and mtDNA replication was also reduced (*Figure 2B–E*). Next, to test whether Myc is sufficient for ETC activity induction, we over-expressed *myc* ORF with a *bam*-Gal4 in the dividing cysts in region 2A that normally have low levels of COX activity and Myc protein. Over-expression of Myc in this region ectopically enhanced COX activity (*Figure 2B,C*). Thus, Myc is both necessary and sufficient to stimulate mitochondrial respiration in the ovary.

To gain insight into how Myc regulates mitochondrial biogenesis, we compared the transcriptomes of wt and $myc^{P0}$ mutant ovaries (*Supplementary file 2*). RNA sequencing (RNAseq) showed that nearly one-third of the detected transcripts were reduced in $myc^{P0}$ mutant compared to wt (fold change >3.0, FDR < 0.05%) (*Figure 2F*), consistent with the notion of Myc as a general transcription activator (*Orian et al., 2003*). We found that the downregulated genes were enriched in nuclear-encoded mitochondrial genes (*Zhang et al., 2019*). About 52% of the total mitochondrial genes, and 75% of ETC genes and factors for mtDNA replication and expression were downregulated (*Figure 2F*, *Supplementary file 3*, and *Supplementary file 4*). Myc directly regulates the expression of its targets by binding to a short sequence, CACGTG (E-box) in the regulatory region (*Kim et al., 2008*). Interestingly, 421 out of 458 down-regulated mitochondrial genes have predicted Myc binding sites in their regulatory regions, further substantiating a role for Myc in promoting mitochondrial biogenesis by boosting the transcription of mitochondrial genes (*Figure 2G* and *Supplementary file 4*). Additionally, 45 transcriptional factors, 33 of which have E-boxes in their regulatory regions, were also decreased in $myc^{P0}$ mutant ovaries (*Figure 2F,G* and *Supplementary file 5*), suggesting that secondary transcriptional controls might also be involved in Myc's regulation of mitochondrial biogenesis.

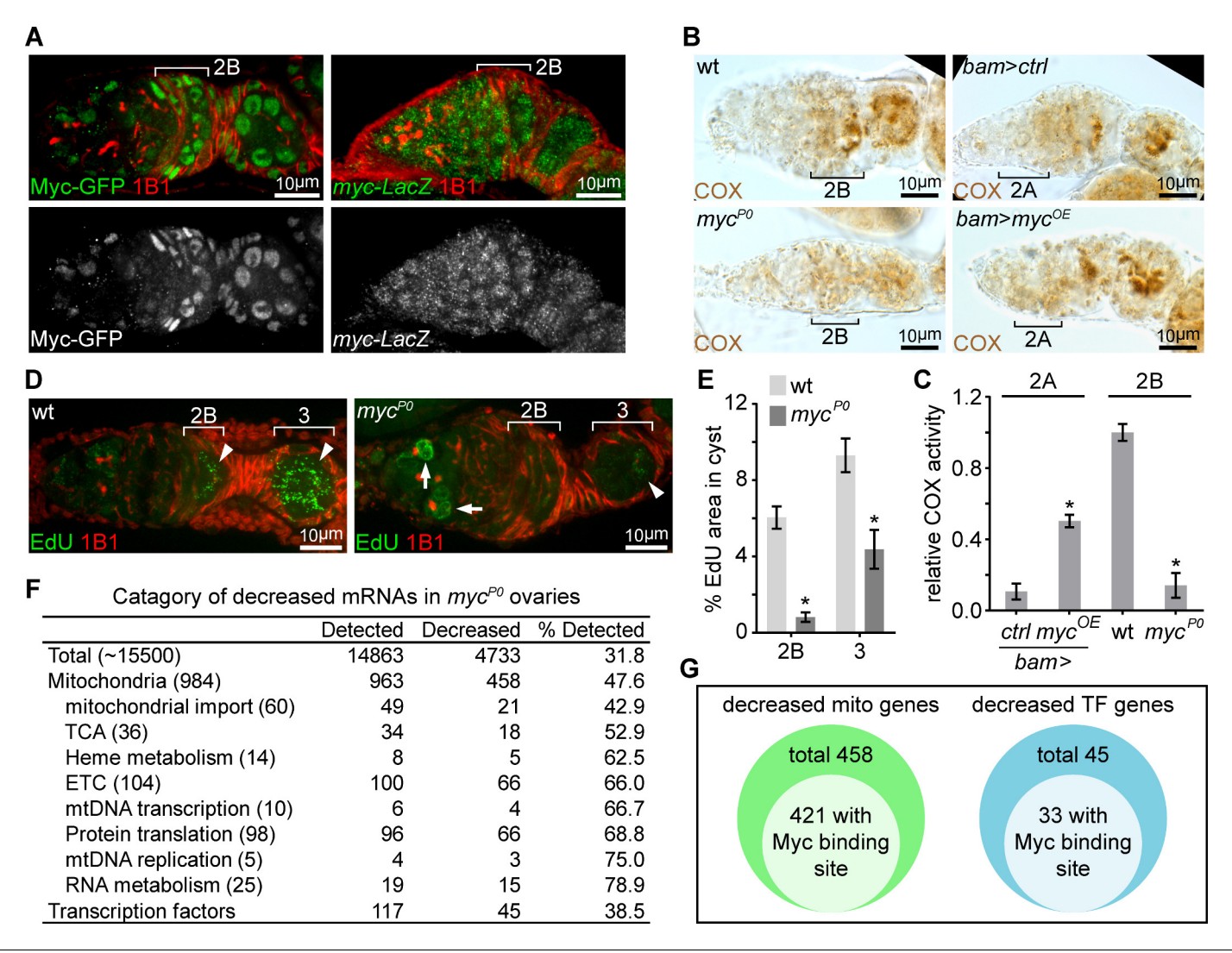

**Figure 2.** A candidate RNAi screen reveals Myc as an essential regulator of mitochondrial biogenesis. (**A**) Left panel: germarium of a fly endogenously expressing Myc-GFP stained with anti-GFP (Green) and anti-1B1 (Red). Right panel: germarium of a fly expressing LacZ driven by the *myc* endogenous promoter stained with anti-β-galactosidase (Green) and anti-1B1 (Red). Myc protein is expressed at low level in GSCs and dividing cysts, but markedly induced from region 2B germ cells. In contrast, *myc* promoter activity is uniform in the germarium. Scale bars, 10 μm. (**B**) COX activity in germaria from wt, $myc^{P0}$, *bam >ctrl*, and *bam >myc^{OE}* ovaries visualized by COX single staining. The activities are normalized to that of region 2B cysts wt. COX activity is significantly reduced in the $myc^{P0}$ mutant, but is ectopically induced when Myc is over-expressed in region 2A by *bam*-Gal4. Scale bars, 10 μm. (**C**) Quantifications of relative COX activity in germarium regions from wt, $myc^{P0}$, *bam >ctrl*, and *bam >myc^{OE}* flies. ETC activities is normalized to that in the wt 2B cysts. n = 10 germaria for each genotype. Error bars represent SEM. *$p<0.05$. (**D**) Visualization of mtDNA replication in germaria from wt and $myc^{P0}$ ovaries with EdU incorporation (Green) and co-staining with anti-1B1 (Red). Arrowheads point to EdU incorporation into mtDNA, while arrows point to EdU incorporation into the nuclear genome. Scale bars, 10 μm. (**E**) Area of EdU puncta (pixels) normalized to total pixels at indicated germarium stages in the germline cysts from wt and $myc^{P0}$ ovaries. n = 11 cysts for each column. Error bars represent SEM. *$p<0.005$. (**F**) Table of genes functioning in mitochondrial processes with at least 3-fold decreased expression in $myc^{P0}$ ovaries compared with wt ovaries. (**G**) Diagrams of decreased genes encoding mitochondrial processes and transcription factors (TFs) in the $myc^{P0}$ ovaries. A number of genes in either category has a Myc binding site in their regulatory region.

DOI: https://doi.org/10.7554/eLife.49309.005

The following source data and figure supplements are available for figure 2:

**Source data 1.** Relative ETC activity and mtDNA area in the germarium regions.
DOI: https://doi.org/10.7554/eLife.49309.008
**Figure supplement 1.** Myc protein pattern in the ovary.
DOI: https://doi.org/10.7554/eLife.49309.006
**Figure supplement 2.** Standards for COX activity staining reveal that staining intensity is linearly correlated with the amount of complexes.

*Figure 2 continued on next page*

*Figure 2 continued*

DOI: https://doi.org/10.7554/eLife.49309.007

## IIS regulates Myc post-transcriptionally through Sgg and Thor

Having identified Myc as the master regulator of ETC biogenesis and mtDNA replication in the ovary, we sought to explore how the spatio-temporal pattern of Myc protein was established. Myc can be regulated either transcriptionally or post-transcriptionally by a variety of upstream signals (*Gallant, 2013*). We first examined *myc* transcription by visualizing its promoter activity using a *myc-LacZ* transgene (*Neto-Silva et al., 2010*). In contrast to Myc protein, which was markedly upregulated at region 2B, *myc* promoter activity appeared to be uniform in the germarium (*Figure 2A*), suggesting that post-transcriptional regulations are responsible for the spatial pattern of Myc protein. IIS/TORC1 signaling is known to regulate both translation and protein stability (*Garofalo, 2002*; *Maurer et al., 2014*; *Pan et al., 2004*), and multiple genes in the IIS/TORC1 signaling emerged from the initial RNAi screen. Consistently, COX activity and mtDNA replication were markedly impaired in ovaries of *chico* mutant flies (*Figure 3A–C*), which were obtained by combining two *chico* mutant alleles, *chico$^1$* (loss of function) and *chico$^{KG}$* (hypomorphic) (*Böhni et al., 1999*; *Song et al., 2010*). These data support a critical role of IIS in ETC biogenesis and mtDNA replication. Intriguingly, the activity of IIS, revealed by staining for phosphorylated AKT at serine 505 (p-AKT) (*Parker and Struhl, 2015*), was also increased in the germarium region 2B and maintained until mid-stage egg chambers (*Figure 3D* and *Figure 3—figure supplement 1A*), a pattern similar to that of Myc protein. In contrast, total AKT staining was uniform in the germarium (*Figure 3D*). These observations suggest that Myc may be regulated by IIS. Indeed, Myc protein was strongly reduced in *chico$^{1/KG}$* mutant ovaries (*Figure 3E,F*). In *chico* RNAi ovaries, Myc protein was also diminished in germ cells, while the expression of *myc-LacZ* was not affected (*Figure 4B* and *Figure 4—figure supplement 1A*). Importantly, over-expressing Myc in the *chico* RNAi background restored ETC biogenesis in the ovary (*Figure 4D*). Altogether, these results suggest that upregulation of IIS in late germarium stimulates ETC biogenesis and mtDNA replication through post-transcriptional control of Myc level.

Next, we explored how IIS regulates Myc. IIS either promotes protein translation by repressing 4E-BP/Thor, or stabilizes its targets by antagonizing GSK3/Sgg-dependent protein degradation (*Figure 4A*) (*Garofalo, 2002*; *Maurer et al., 2014*; *Pan et al., 2004*). Knocking down *sgg* in a *chico* RNAi background restored Myc protein level and COX activity, while *thor* RNAi only partially rescued both (*Figure 4B–D*). Intriguingly, *sgg* RNAi not only elevated Myc protein level in region 2B and thereafter, but also strongly induced Myc in earlier stages where Myc protein is not normally present (*Figure 4E,F*). Next, we examined the pattern of GSK3 and GSK3 activity in the germarium. GSK3 activity is suppressed by IIS through AKT-mediated phosphorylation on GSK3 serine 9. GSK3 protein visualized by both antibody staining and an endogenous expressed Sgg-GFP was ubiquitous in the germarium (*Figure 4G* and *Figure 4—figure supplement 1B*). However, phosphorylated GSK3 (*Figure 4G* and *Figure 4—figure supplement 1C*), the inactive form of GSK3, became evident in region 2B germarium and later stages egg chambers, the same pattern as ETC biogenesis (*Figure 1B,C*), Myc (*Figure 2B* and *Figure 4—figure supplement 1C*), and p-AKT (*Figure 3D*). Taken together, these data suggest that Sgg is the main regulator of Myc and acts downstream of IIS.

## InR expression is boosted at region 2B germarium

So far, our data has established Myc as the link between IIS, a major pathway regulating cell proliferation and growth, and mitochondrial biogenesis in ovaries. The IIS regulates germ cells growth and proliferation in response to insulin-like peptides (dilps) produced by neuroendocrine cells (*LaFever and Drummond-Barbosa, 2005*). *Drosophila* has an open circulatory system. In a given tissue, all cells are exposed to a similar level of dilps circulating in the hemolymph. However, instead of being uniform in the germarium, the activity of IIS, indicated by both p-AKT and inhibitory phosphorylated Sgg staining (*Figures 3D* and *4G*), demonstrated a distinct spatio-temporal pattern similar to that of Myc, ETC expression and mtDNA replication. Therefore, some IIS components downstream of dilps must be differentially expressed in the germarium.

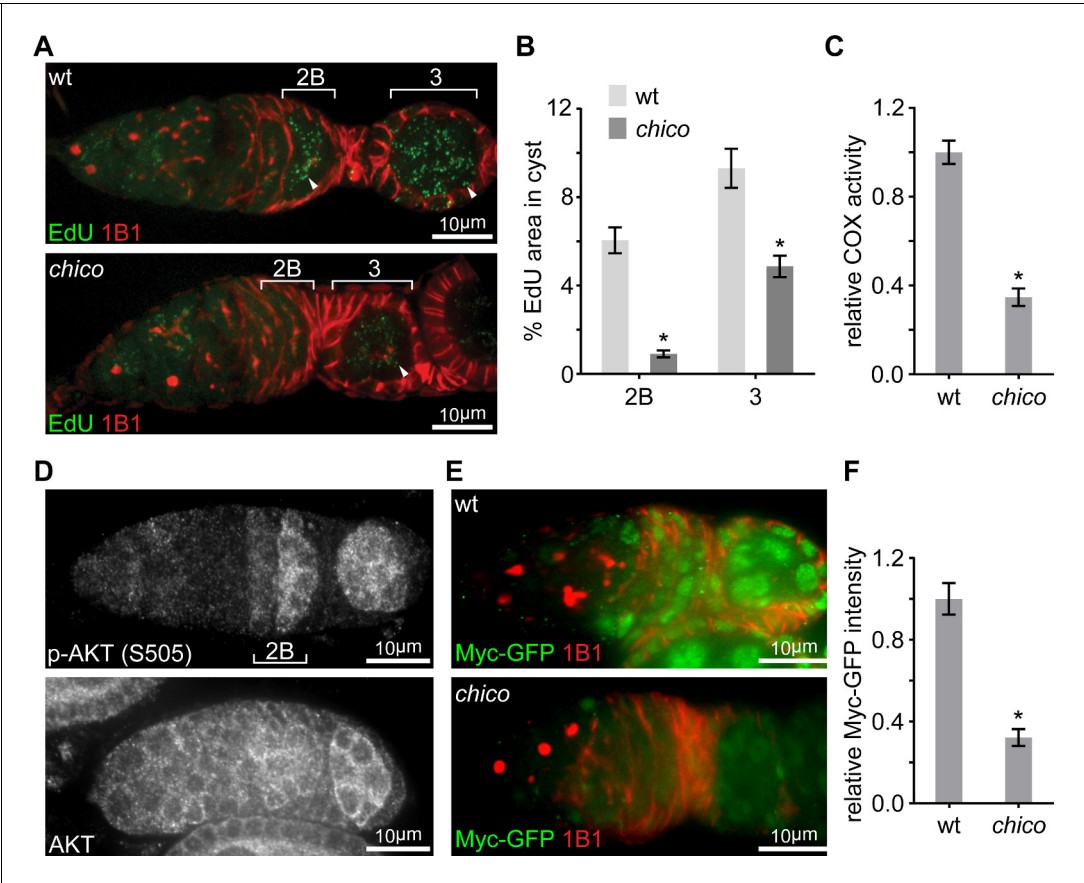

**Figure 3.** Elevated IIS in late germarium induces Myc protein to prime mtDNA replication and mitochondrial respiration. (**A**) Representative germaria from wt or *chico*[1/KG] mutant flies incorporating EdU and stained with anti-1B1. Arrowheads indicate EdU incorporation into the mtDNA of germ cells. Scale bars, 10 μm. (**B**) Quantification of the mitochondrial EdU puncta areas relative to the areas of germline cysts from wt or *chico*[1/KG] mutant flies. n = 11 cysts each column (left to right). Error bars represent SEM. *$p<0.005$. (**C**) Quantifications of COX activity in region 2B cysts from wt or *chico*[1/KG] mutant flies. The activities are normalized to that of region 2B cysts from wt. n = 10 germaria for each genotype. Error bars represent SEM. *$p<0.005$. (**D**) Germaria from wt ovaries stained with anti-AKT and anti-p-AKT (S505). p-AKT staining is low in both GSCs and dividing cysts, while increased from region 2B germ cells. In contrast, AKT staining is uniform in the germarium. Scale bars, 10 μm. (**E**) Germaria from wt or *chico*[1/KG] mutant ovaries endogenously expressing Myc-GFP stained with anti-GFP, anti-1B1, and DAPI. Scale bars, 10 μm. (**F**) Quantification of relative Myc-GFP intensity in germarium region 2B from wt or *chico*[1/KG] mutant ovaries. n = 10 germaria for each genotype. Error bars represent SEM. *$p<0.005$.
DOI: https://doi.org/10.7554/eLife.49309.009

The following source data and figure supplement are available for figure 3:

**Source data 1.** Relative mtDNA area, COX activity and Myc-GFP intensity in the germarium.
DOI: https://doi.org/10.7554/eLife.49309.011

**Figure supplement 1.** IIS activity in the ovary.
DOI: https://doi.org/10.7554/eLife.49309.010

To test this idea, we generated an InR-EGFP reporter line by inserting an EGFP at the C-terminus of the *InR* genomic locus. Using this line and other reporter lines (*Nagarkar-Jaiswal et al., 2015*; *Orme et al., 2006*; *Sarov et al., 2016*), we examined the expression patterns of InR and other components of IIS signaling upstream of AKT. InR-EGFP was upregulated in region 2B (*Figure 4H*), while all other components in the IIS signaling examined were ubiquitously expressed in the germarium (*Figure 5—figure supplement 1A*). Additionally, *InR* mRNA, visualized by FISH, demonstrated the same pattern as that of InR-EGFP (*Figure 4H*), suggesting that upregulation of *InR* transcription enhances IIS to boost mitochondrial biogenesis.

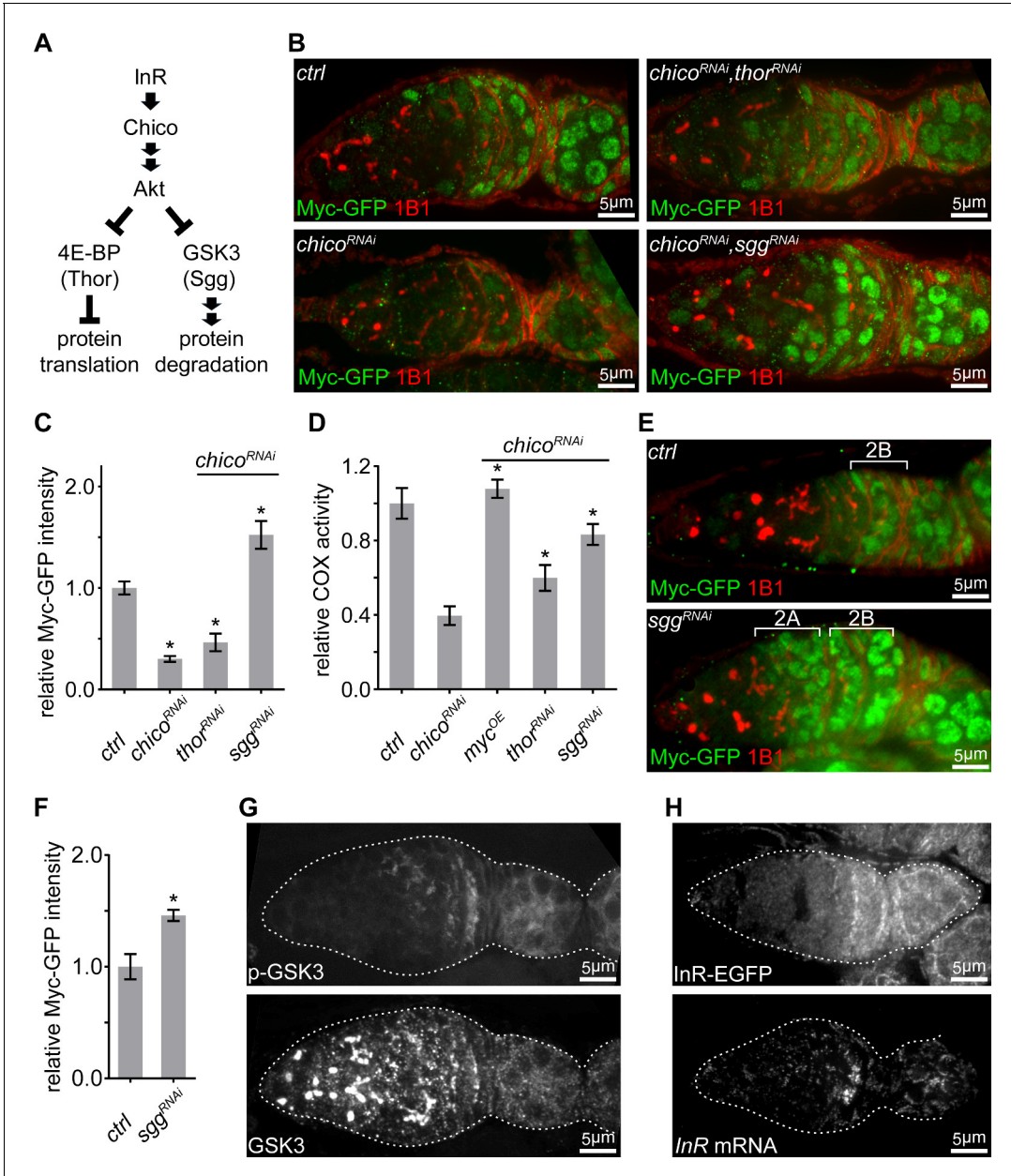

**Figure 4.** IIS promotes ETC activity via inhibition of GSK3 to stabilize Myc. (**A**) Schematic of the conserved IIS pathway that inhibits 4E-BP/Thor and GSK3/Sgg to promote protein translation and suppress protein degradation, respectively. (**B**) Germaria from ovaries of *ctrl*; *chico^RNAi^*; *chico^RNAi^*, *thor^RNAi^*; and *chico^RNAi^*, *sgg^RNAi^* driven by *nos*-Gal4 endogenously expressing Myc-GFP and stained with anti-GFP (green) and anti-1B1 (red). Scale bars, 5 μm. (**C, D**) Quantification of Myc-GFP intensity (**C**) and COX activity (**D**) in germarium region 2B from ovaries with indicated genotypes, normalized to the intensity or activity values in germaria with *ctrl* over-expression. (**C**) n = 10 germaria for each genotype. (**D**) n = 10 germaria for each genotype. Error bars represent SEM. *p<0.05. (**E**) Germaria from ovaries of *ctrl* and *sgg^RNAi^* driven by *nos*-Gal4 endogenously expressing Myc-GFP and stained with anti-GFP and anti-1B1. Myc protein is up-regulated in both region 2A and region 2B germ cells in the *sgg* RNAi ovary. Scale bars, 10 μm. (**F**) Quantification of relative Myc-GFP intensity in germarium region 2B from ovaries of *ctrl* and *sgg^RNAi^* driven by *nos*-Gal4. Myc-GFP intensity is normalized to that of region 2B cysts with *ctrl* expression. n = 7 germaria for each genotype. Error bars represent SEM. *p<0.01. (**G**) A germarium from wt flies stained with anti-GSK3 and anti-p-GSK3. Germaria are outlined with dotted lines. Scale bars, 5 μm. (**H**) Upper panel: a representative image of a germarium from ovaries expressing endogenous InR-EGFP. Lower panel: visualization of the *InR* mRNA in germarium by FISH with fluorescently labeled DNA probes. Germaria are outlined with dotted lines. Scale bars, 5 μm.

DOI: https://doi.org/10.7554/eLife.49309.012

The following source data and figure supplement are available for figure 4:

**Source data 1.** Relative COX activity and Myc-GFP intensity in the germarium.

*Figure 4 continued on next page*

*Figure 4 continued*

DOI: https://doi.org/10.7554/eLife.49309.014

**Figure supplement 1.** Myc is post-transcriptionally regulated by the IIS-GSK3 cascade.

DOI: https://doi.org/10.7554/eLife.49309.013

## Both JNK pathway and Myc promote IIS via *InR* transcription

We next asked how *InR* transcription was elevated at region 2B germarium. The JNK pathway, which transcriptionally controls various cellular processes (*Weston and Davis, 2007*), had emerged from the initial RNAi screen. Consistent with the RNAi screen, homozygous *bsk¹* clones showed impaired mtDNA replication, compared to *bsk¹/+* heterozygous germ cells (*Figure 5A,B*). Interestingly, JNK signaling activity, visualized by a *puc-LacZ* reporter (*Martín-Blanco et al., 1998*), was sharply up-regulated in late germarium stages, but decreased and eventually disappeared in growing egg chambers (*Figure 5C* and *Figure 5—figure supplement 1B*). The partial overlap between the spatial patterns of IIS and JNK activation, and the phenotypic resemblance between IIS and JNK mutations on COX activity and mtDNA replication, suggested a potential link between these two pathways. Indeed, IIS activity, Myc protein, InR-EGFP, and *InR* mRNA were all markedly reduced in ovaries expressing dsRNA against either *bsk* or *jra*, fly homolog of JNK or Jun, respectively (*Figure 5D–H* and *Figure 5—figure supplement 1C–G*). In *bsk* RNAi or *jra* RNAi background, over-expression of *InR* restored Myc level and over-expression of either *InR* or *Myc* rescued COX activity (*Figure 5H,I* and *Figure 5—figure supplement 1G*). In contrast, enhancing JNK signaling by *puc* RNAi failed to rescue defective ETC activity in a *chico* RNAi background (*Figure 5I*). Together, these observations suggest that JNK promotes mitochondrial biogenesis respiration and mtDNA replication in late germarium stages by boosting IIS.

IIS was elevated in region 2B and remained active until stage-10 egg chambers, the same period during which ETC biogenesis and mtDNA replication are active. However, JNK is only transiently activated in the region 2B cysts. Therefore, additional regulations must be involved to maintain IIS activity after JNK activity subsides. Our RNAseq results showed that *InR* mRNA was downregulated in the *myc^{P0}* ovary compared with controls (*Supplementary file 2*), suggesting that Myc might activate *InR* transcription. Indeed, both *InR* mRNA level measured by FISH and IIS activity indicated by p-AKT were reduced in *myc^{P0}* mutant or *chico* RNAi ovaries (*Figure 6A–D* and *Figure 6—figure supplement 1A,B*). Importantly, over-expression of *myc* using *bam*-Gal4 in region 2A ectopically induced *InR* transcription and IIS activity (*Figure 6A–D* and *Figure 6—figure supplement 1A,B*), suggesting that Myc can indeed increase IIS activity by boosting *InR* expression. Together, our results highlight an IIS-Myc-positive feedback loop that promotes respiration and mtDNA replication in the ovary.

## The JNK-IIS-Myc relay is essential for female fertility and mtDNA selective inheritance

So far, we have established that the JNK-IIS-Myc relay is critical for ETC activity, mtDNA expression, replication, and transmission in the ovary. Next, we explored its physiological impact on reproduction. While both *chico* mutant females and those with germline clones of *bsk¹* produced similar amount of eggs as controls, they failed to generate adequate amount of mtDNA to deposit in eggs (*Figure 7A*). Thus, their eggs had significantly reduced mtDNA level and hatching rates (*Figure 7B*).

During oogenesis, prodigious mtDNA replication not only furnishes mature oocytes with adequate amounts of mtDNA to support the early embryogenesis, it also affords the replication competition that allows the wild-type mitochondrial genomes to out-compete mtDNA carrying deleterious mutations (*Hill et al., 2014*). Thus, we asked whether inhibition of IIS and JNK signaling, which impairs mtDNA replication, would also diminish selective inheritance in heteroplasmic females harboring both wt and a temperature-sensitive lethal mutation, *mt:CoI^{T300I}* (*Hill et al., 2014*). Consistent with previous studies, eggs contained ~20% less *mt:CoI^{T300I}* mtDNA on average than their mothers at restrictive temperature in controls (*Figure 7C*). However, this counter-selection of the *mt:CoI^{T300I}* genome was greatly diminished by downregulation of either IIS or JNK signaling (*Figure 7C*). Together, these results stress that although JNK is transiently activated in the late germarium, it triggers a developmental signaling relay that has profound impacts on mitochondrial

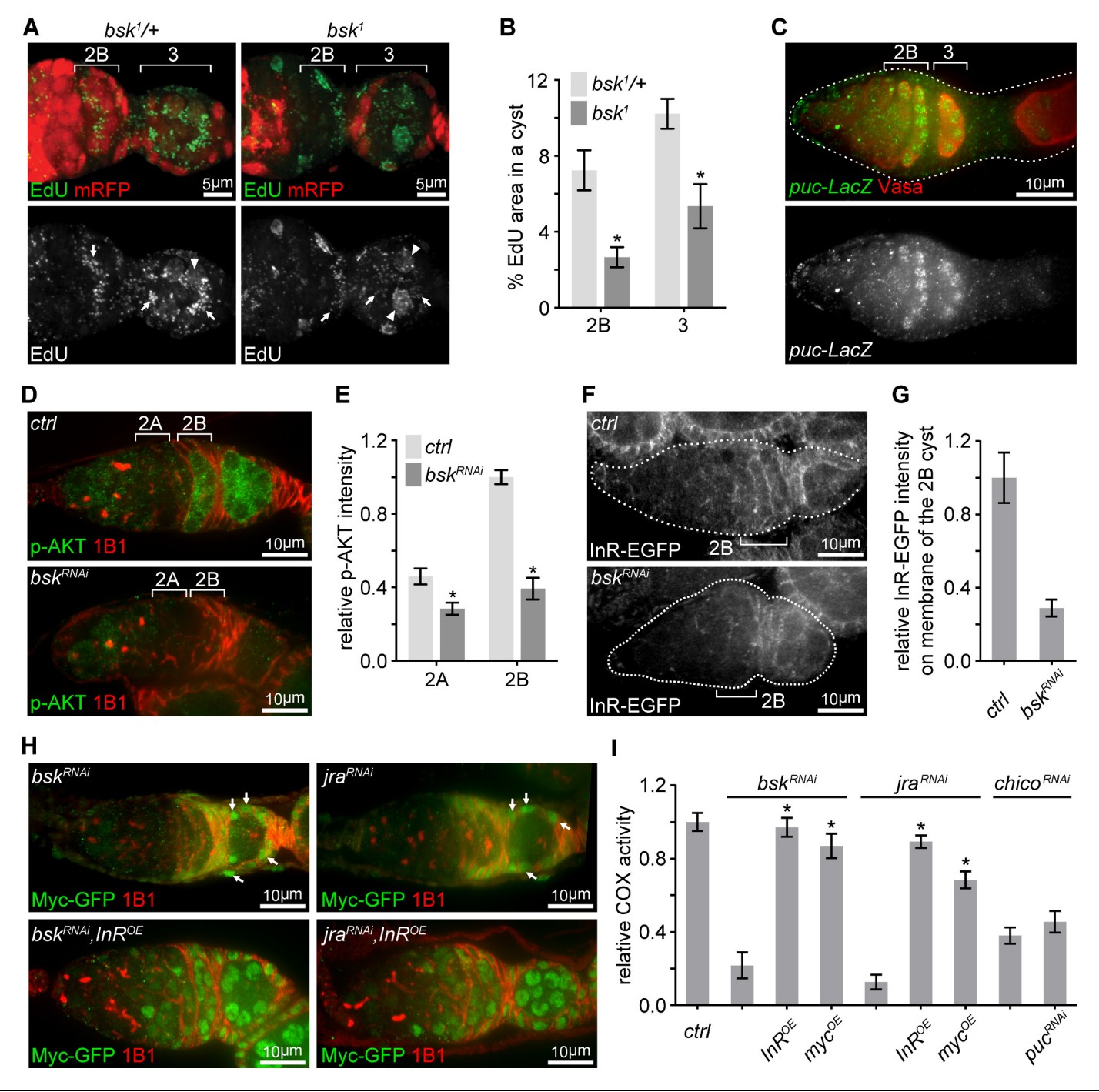

**Figure 5.** Transient JNK activation in the germarium boosts the IIS-Myc signaling. (**A**) Representative germaria with *bsk¹* FRT clones showing EdU incorporation (green) to visualize mtDNA replication. *bsk¹/+* cells are positive for mRFP (red), while *bsk¹* mutant cells are negative for mRFP. Arrows indicate EdU incorporated into mtDNA, arrowheads point out EdU incorporated into the nuclear genome. Scale bars, 5 μm. (**B**) Percentage areas of EdU incorporated into mtDNA relative to total areas of the germline cysts at indicated germarium stages from *bsk¹/+* and *bsk¹* clones. n = 11, 14, 15, and 10 cysts each column (left to right). Error bars represent SEM. *p<0.005. (**C**) Germarium from ovaries expressing *LacZ* driven by the *puc* promoter stained with anti-β-galactosidase (green) and anti-Vasa (red). Germarium is outlined with dotted lines. Scale bar, 10 μm. (**D**) Germaria from ovaries of *ctrl* and *bsk^RNAi* driven by *nos*-Gal4 stained with anti-p-AKT (green) and anti-1B1 (red). Note that IIS activity is markedly reduced when JNK signaling is decreased by *bsk* RNAi. Scale bars, 10 μm. (**E**) Quantification of p-AKT intensity from cysts in germarium region 2A and 2B of ovaries with indicated genotypes. p-AKT intensity is normalized to that of region 2B cysts from the *ctrl* line. n = 11 germaria for each genotype. Error bars represent SEM. *p<0.005. (**F**) Visualization of InR-EGFP in germaria from ovaries of *ctrl* and *bsk^RNAi* driven by *nos*-Gal4. Note that InR-EGFP level on the region 2B cyst

*Figure 5 continued on next page*

*Figure 5 continued*

membrane is decreased by *bsk* RNAi. Germaria are outlined with dotted lines. Scale bars, 10 µm. (G) Quantification of background subtracted InR-EGFP intensity on cell membrane of the region 2B cyst. n = 7 germaria for each genotype. Error bars represent SEM. *p<0.005. (H) Germaria from ovaries of $bsk^{RNAi}$; $bsk^{RNAi}$, $InR^{OE}$; $jra^{RNAi}$; $jra^{RNAi}$, $InR^{OE}$ driven by *nos*-Gal4 endogenously expressing Myc-GFP co-stained with anti-GFP and anti-1B1. Note that Myc-GFP level in germ cells is markedly lower than that in follicle cells pointed out by arrows. Scale bars, 10 µm. (I) COX activity in region 2B cysts from ovaries with indicated genotypes. The activities are normalized to that of region 2B cysts from the *ctrl* line. n = 10 germaria for each genotype. Error bars represent SEM. *p<0.005.

DOI: https://doi.org/10.7554/eLife.49309.015

The following source data and figure supplements are available for figure 5:

**Source data 1.** Relative mtDNA area, p-AKT intensity, InR-EGFP intensity, and COX activity in the germarium.

DOI: https://doi.org/10.7554/eLife.49309.018

**Figure supplement 1.** Other IIS components are not altered by *bsk* RNAi.

DOI: https://doi.org/10.7554/eLife.49309.016

**Figure supplement 1—source data 1.** Relative AKT intensity, *InR* mRNA density, and Myc-GFP intensity in the germarium.

DOI: https://doi.org/10.7554/eLife.49309.017

inheritance through activation of mitochondrial biogenesis, including ETC biogenesis and mtDNA replication.

## Discussion

mtDNA replication in the *Drosophila* ovary relies on active respiration (*Hill et al., 2014*), suggesting that ETC activity and mtDNA replication might be subject to the same spatio-temporal regulation. In this study, we address this question and further elucidate the developmental mechanisms regulating ETC activity and mtDNA biogenesis in the ovary. Utilizing the COX/SDH dual activity staining, we reveal that ETC complexes are inactive in the germline stem cells (GSCs) and dividing cysts from germarium region 1 to 2A, but sharply activated in region 2B and active through stage-10 follicles. This spatial pattern mirrors that of mtDNA replication in the *Drosophila* ovary, supporting an essential role of mitochondrial respiration in mtDNA inheritance, both quantitively and qualitatively. We also demonstrate that ETC activation is accompanied with an upregulation of the expression of ETC genes of both nuclear and mitochondrial origin. Interestingly, MDI, which drives the local translation of nuclear encoded mitochondrial proteins on the mitochondrial outer membrane and TFAM, which governs mtDNA replication and transcription (*Falkenberg et al., 2007*; *Zhang et al., 2016*), exhibit the same developmental pattern as mitochondrial respiration in the germarium. Collectively, these proteins would boost the biogenesis of ETC in region 2B of the germarium and in growing egg chambers. In an ovariole, different stages of developing germ cells reside in the same microenvironment and experience the same oxygen tension. Thus, the mitochondrial respiratory activity is likely to be determined by the abundance of ETC components, which itself is controlled by transcriptional activation.

To understand how mitochondrial respiration is regulated, we conducted an RNAi screen for genes that boost COX/SDH activity in the ovary. The *myc* gene emerged as one of the strongest hits, and a hypomorphic allele, $myc^{P0}$, largely abolished ETC activity and mtDNA replication in the germarium. Moreover, the spatial pattern of Myc protein mirrors mtDNA replication and ETC activity, further supporting its essential role in transcriptional activation of ETC biogenesis. RNA sequencing data demonstrate that Myc broadly stimulates gene expression in the *Drosophila* ovary, including many nuclear-encoded ETC genes and factors required for mtDNA replication and expression. Our observations are consistent with previous studies in mammals showing that MYC can promote mitochondrial biogenesis by directly elevating the expression of nuclear-encoded mitochondrial genes (*Kim et al., 2008*; *Li et al., 2005*; *Stine et al., 2015*). Among 198 annotated human mitochondrial genes that are up-regulated by Myc overexpression (*Li et al., 2005*), 185 have homologs in the *Drosophila* genome (*Supplementary file 6*). Of note, 44.9% (101 out of 225) of the fly homologs are down-regulated in $myc^{P0}$ mutant ovaries (*Supplementary file 6*), suggesting an evolutionarily conserved function of Myc in regulating mitochondrial biogenesis through gene expression. Our finding that Myc induces ETC biogenesis and respiration is also in line with the studies in mammals demonstrating the multi-faceted roles of Myc in the regulation of mitochondria,

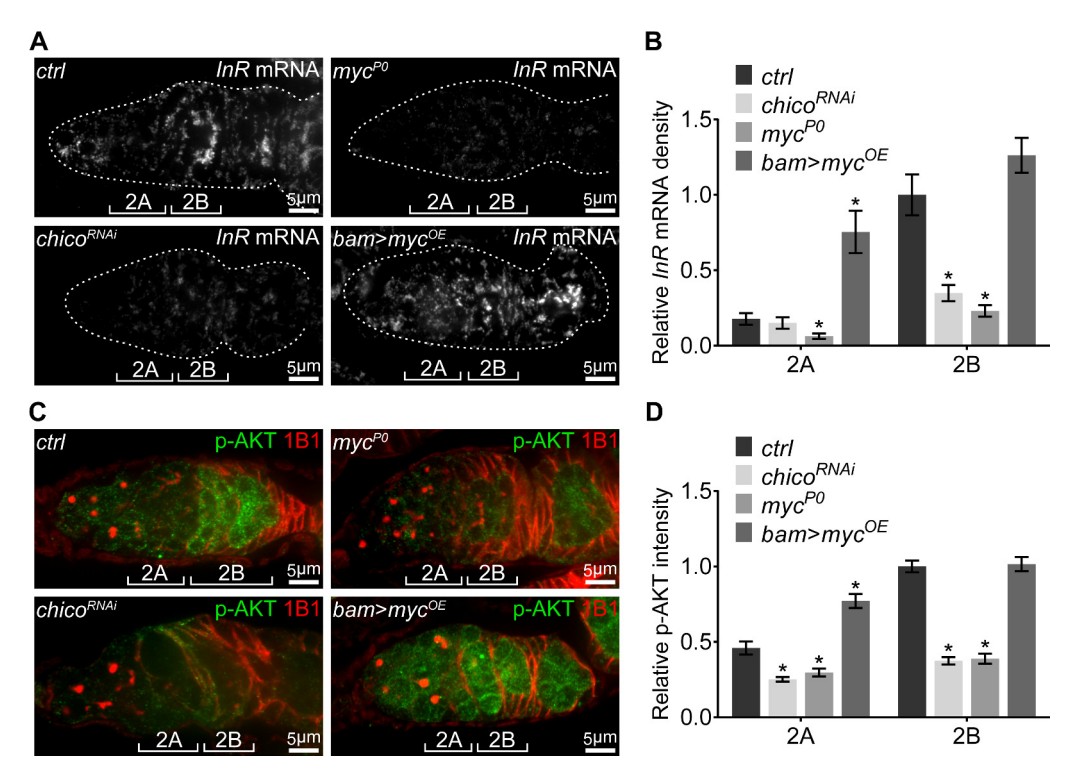

**Figure 6.** A positive feedback regulatory loop between IIS and Myc. (**A**) Visualization of the *InR* mRNA by FISH with fluorescently labeled DNA probes in germaria from *ctrl*, *chico^RNAi*, *myc^P0*, and *bam >myc^OE* ovaries. Germaria are outlined with dotted lines. Reduction in either IIS or Myc depletes *InR* mRNA in the germarium, while *myc* over-expression in region 2A ectopically induces *InR* mRNA. Scale bars, 5 μm. (**B**) Quantification of *InR* mRNA intensity from cysts in germarium region 2A and 2B of ovaries with indicated genotypes. Intensities are normalized to the value of *ctrl* at region 2B. n = 8, 8, 8, and 9 germaria for *ctrl*, *chico^RNAi*, *myc^P0*, and *bam >myc^OE*, respectively. Error bars represent SEM. *p<0.05. (**C**) Germaria from *ctrl*, *chico^RNAi*, *myc^P0*, and *bam >myc^OE* ovaries stained with anti-p-AKT and anti-1B1. Decrease in either IIS or Myc reduces IIS activity in the germarium, while *myc* over-expression in region 2A ectopically induces IIS activity. Scale bars, 5 μm. (**D**) Quantification of p-AKT intensity in region 2A and 2B cysts of ovaries with indicated genotypes. Intensities are normalized to the value of *ctrl* at region 2B. n = 11 germaria for each genotype. Error bars represent SEM. *p<0.005.

DOI: https://doi.org/10.7554/eLife.49309.019

The following source data and figure supplements are available for figure 6:

**Source data 1.** Relative *InR* mRNA density and p-AKT intensity in the germarium.

DOI: https://doi.org/10.7554/eLife.49309.022

**Figure supplement 1.** AKT level is normal when IIS is decreased, but slightly reduced in the *myc^P0* mutant.

DOI: https://doi.org/10.7554/eLife.49309.020

**Figure supplement 1—source data 1.** Relative AKT intensity in the germarium.

DOI: https://doi.org/10.7554/eLife.49309.021

including boosting mitochondrial biogenesis (*Ahuja et al., 2010*; *Kim et al., 2008*; *Stine et al., 2015*), stimulating oxidative metabolism (*Ahuja et al., 2010*), and regulating mitochondrial structure and dynamics (*Graves et al., 2012*).

Myc overexpression sometimes gives rise to different transcriptional output in different cell types (*de la Cova et al., 2014*). This observation reflects the fact that Myc-family proteins often associate with other cofactors and exert a broad and complex transcriptional role in a cell- or tissue-specific manner (*Cowling and Cole, 2006*; *Hann, 2014*). We also found that 130 transcription regulators, including *Srl* (fly homolog of human PGC-1) and CG32343 (fly homolog of GABPB2), were affected by the *myc^P0* mutation. PGC-1 proteins belong to an evolutionarily conserved family that integrates mitochondrial biogenesis and energy metabolism with a variety of cellular processes (*Lin et al., 2005*). In *Drosophila*, *Srl* regulates the expression of a subset of nuclear encoded mitochondrial genes (*Tiefenböck et al., 2010*). Mammalian GABPB2 is a regulatory subunit of the Nuclear

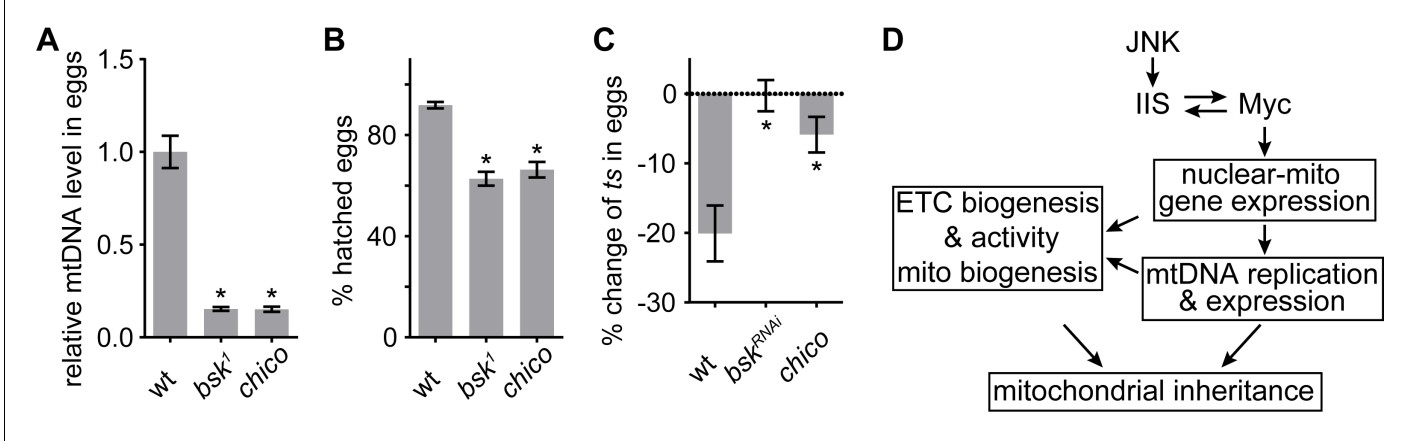

**Figure 7.** JNK-IIS relay is essential for mtDNA selective inheritance and fertility. (**A**) Quantification of relative mtDNA content in eggs produced by mothers carrying germline clones (GLCs) of $bsk^1$ or by $chico^{1/KG}$ mutant mother. Relative mtDNA levels are determined by qPCR for $mt:CoI$ and $his4$ copies, and normalized to the level in wt. n = 12, 16, and 12 mothers for wt, $bsk^1$, and $chico^{1/KG}$. Error bars represent SEM. *p<0.005. (**B**) The hatching rate of eggs produced by female flies carrying $bsk^1$ GLCs or $chico^{1/KG}$ females is significantly lower than that of wt eggs. n = 5 × 40 eggs for each genotype. Error bars represent SEM. *p<0.005. (**C**) Quantification of $mt:CoI^{T300I}$ transmission from females with indicated nuclear genotypes. In wt females, the mtDNA mutation is counter-selected, resulting in ~20% fewer mutant mtDNA in the progeny than in the mothers. This counter-selection is diminished in $chico^{1/KG}$ mutant mothers of in mothers expressing $bsk^{RNAi}$ driven by $nos$-Gal4. Mothers heteroplasmic for $mt:CoI^{T300I}$ were cultured at 29°C. n = 10 mothers for each genotype. Error bars represent SEM. *p<0.005. (**D**) Schematics of the developmental signaling relay initiated from late germarium that primes mitochondrial respiration, and mtDNA replication and inheritance.

DOI: https://doi.org/10.7554/eLife.49309.023

The following source data is available for figure 7:

**Source data 1.** Relative mtDNA level in eggs, % hatched eggs, and % change of $ts$ in eggs.

DOI: https://doi.org/10.7554/eLife.49309.024

Respiratory Factor complex 2 that regulates the expression of a small set of nuclear encoded mitochondrial proteins (*Kelly and Scarpulla, 2004*). Therefore, additional tiers of transcriptional regulations downstream of Myc are likely involved in boosting ETC biogenesis.

While *myc* transcription is uniform in the germarium, Myc protein is elevated at region 2B and remains high until the stage-10 egg chamber, indicating that Myc abundance is mainly regulated *via* post transcriptional mechanisms. IIS and JNK also emerged from our RNAi screen, and both were further confirmed to be required for triggering ETC biogenesis and mtDNA replication. We found that IIS activity, marked by both p-AKT and p-GSK3 staining, displayed a pattern similar to that of Myc. Additionally, elevated IIS activity was required to establish a high level of Myc and to activate ETC in the late germarium stage. GSK3 directly phosphorylates Myc and promotes its ubiquitination and degradation in both mammalian and fly cultured cells (*Galletti et al., 2009*; *Jellusova et al., 2017*; *Sears et al., 2000*). Thus, IIS likely stabilizes Myc protein by inhibiting GSK activity. Our result is also in line with a previous study showing that decreased IIS activity relieves the inhibition on GSK3, which leads to mitochondrial quiescence at later stages of oogenesis (*Sieber et al., 2016*). Importantly, our work uncovers Myc as the downstream effector of IIS in the regulation of respiration and mtDNA biogenesis in the ovary.

We noticed that *InR* transcription was down-regulated in the *myc* mutant ovary, suggesting a positive feedback regulation between IIS and Myc. This regulatory loop maintains high levels of both Myc protein and IIS activity in the mid-stage follicles, where massive mitochondrial biogenesis and massive cell growth take place. However, it does not explain how this loop is activated in the first place at the late germarium stages. We found that JNK was transiently activated in germ cells in the germarium region 2B, but decreased in budding egg chambers and sharply diminished thereafter. High level and sustained JNK activity often lead to apoptosis. However, cell death is rarely observed in the germaria of flies cultured under normal conditions. Thus, JNK activation in the late germarium must be triggered by cellular processes unrelated to apoptosis. We reveal that transiently elevated JNK activity is sufficient to increase *InR* mRNA level, which in-turn boosts IIS activity and stabilizes

Myc protein. Currently, the link between JNK and IIS is not well-understood. In the metastatic *Drosophila* epithelium, cell survival and proliferation entail upregulation of *InR* expression by JNK through wingless signaling (*Hirabayashi et al., 2013*). However, no genes in the wingless signaling pathway emerged from our RNAi screen in germ cells. The molecular mechanisms that links JNK activation to *InR* expression in ovary remain to be explored.

The JNK-dependent transcriptional program can be activated by various cellular stresses and cell-cell signaling events (*Ríos-Barrera and Riesgo-Escovar, 2013*). In region 2B of the germarium, the follicle cells extend and migrate laterally across the germarium to wrap around the 16 cells cyst (*Nystul and Spradling, 2007*). Thus, JNK activation in germ cells may reflect paracrine signaling from the follicle cells, for instance via TNF-α. Alternatively, the process of follicle cells enveloping and compressing the 16-cell cyst may generate mechanical stress that subsequently activates JNK. Regardless, our work uncovers a novel function of JNK in energy metabolism and mitochondrial biogenesis besides its well-established roles in controlling cell apoptosis, growth, and proliferation.

Studies in a variety of animal models have shown that reproductive aging in females is tightly associated with decreased IIS activity (*Templeman and Murphy, 2018*). Interestingly, oocytes of aged females often have higher incidence of mtDNA lesions and lower mtDNA copy number (*Chan et al., 2005*). Thus, developmental control of mitochondrial biogenesis and mtDNA replication via IIS may be a conserved mechanism in metazoans. Our previous studies demonstrated that prodigious mitochondrial biogenesis during oogenesis underlies the selective inheritance of functional mtDNA by allowing proliferation competition between healthy mitochondria and mitochondria carrying deleterious mtDNA mutations (*Zhang et al., 2019*). Here, we uncover that the JNK/IIS/Myc signaling relay governs mitochondrial biogenesis in the ovary, and thereby influences mitochondrial inheritance both quantitively and quantitively. Our studies could provide a molecular framework to further understand the control of mitochondrial biogenesis and mtDNA inheritance in animals.

# Materials and methods

## Key resources table

| Reagent type (species) or resource | Designation | Source or reference | Identifiers | Additional information |
|---|---|---|---|---|
| Genetic reagent (*D. melanogaster*) | $w^{1118}$ (wt) | *Zhang et al., 2019* | https://doi.org/10.1016/j.molcel.2019.01.013 | |
| Genetic reagent (*D. melanogaster*) | ts ($mt:Col^{T300I}\%$) | *Hill et al., 2014* | https://doi.org/10.1038/ng.2920 | |
| Genetic reagent (*D. melanogaster*) | UAS-Dcr-2; nos-Gal4 | Bloomington Drosophila Stock Center | BDSC: 25751 FLYB: FBti0101430 FLYB: FBti0012287 | FLYB symbol: P{UAS-Dcr-2.D} 1 P{GAL4-nos.NGT}40 |
| Genetic reagent (*D. melanogaster*) | nos-Gal4 | Bloomington Drosophila Stock Center | BDSC: 32563 FLYB: FBti0012287 | FLYB symbol: P{GAL4-nos.NGT}40 |
| Genetic reagent (*D. melanogaster*) | bam-Gal4 | *Chen and McKearin, 2003* | https://doi.org/10.1242/dev.00325 | |
| Genetic reagent (*D. melanogaster*) | UAS-LacZ | *Wang et al., 2016* | https://doi.org/10.1093/hmg/ddw067 | *ctrl* over-expression |
| Genetic reagent (*D. melanogaster*) | $myc^{P0}$ | *Johnston et al., 1999* | https://doi.org/10.1016/S0092-8674(00)81512–3 | |
| Genetic reagent (*D. melanogaster*) | myc-LacZ | Bloomington Drosophila Stock Center | BDSC: 12247 FLYB: FBti0015660 | FLYB symbol: P{lacW}Myc$^{G0359}$ |
| Genetic reagent (*D. melanogaster*) | puc-LacZ | Bloomington Drosophila Stock Center | BDSC: 11173 FLYB: FBti0005134 | FLYB symbol: P{lArB}puc$^{A251.1F3}$ |
| Genetic reagent (*D. melanogaster*) | UAS-myc | (*Rhiner et al., 2009*) | https://doi.org/10.1242/dev.033340 | |

*Continued on next page*

*Continued*

| Reagent type (species) or resource | Designation | Source or reference | Identifiers | Additional information |
|---|---|---|---|---|
| Genetic reagent (*D. melanogaster*) | *bsk[1]*, FRT40A | (*Bornstein et al., 2015*) | https://doi.org/10.1016/j.neuron.2015.10.023 | |
| Genetic reagent (*D. melanogaster*) | hs-FLP; ubi-mRFP, FRT40A | (*Bornstein et al., 2015*) | https://doi.org/10.1016/j.neuron.2015.10.023 | |
| Genetic reagent (*D. melanogaster*) | *luciferase[RNAi]* | Bloomington Drosophila Stock Center | BDSC: 31603 FLYB: FBti0130444 | FLYB symbol: P{TRiP.JF01355} attP2 ctrl RNAi |
| Genetic reagent (*D. melanogaster*) | *sgg[RNAi]* | Bloomington Drosophila Stock Center | BDSC: 35364 FLYB: FBst0035364 | FLYB symbol: P{TRiP.GL00277}attP2 |
| Genetic reagent (*D. melanogaster*) | *thor[RNAi]* | Bloomington Drosophila Stock Center | BDSC: 36815 FLYB: FBst0036815 | FLYB symbol: P{TRiP.GL01034} attP2 |
| Genetic reagent (*D. melanogaster*) | *puc[RNAi]* | Bloomington Drosophila Stock Center | BDSC: 36085 FLYB: FBst0036085 | FLYB symbol: P{TRiP.GL00504} attP40 |
| Genetic reagent (*D. melanogaster*) | UAS-InR | Bloomington Drosophila Stock Center | BDSC: 8262 FLYB: FBst0008262 | FLYB symbol: P{UAS-InR.Exel}2 |
| Genetic reagent (*D. melanogaster*) | *chico[1]* | Bloomington Drosophila Stock Center | BDSC: 10738 FLYB: FBst0010738 | FLYB symbol: P{ry11}chico$^{1}$/CyO |
| Genetic reagent (*D. melanogaster*) | *chico[KG]* | Bloomington Drosophila Stock Center | BDSC: 14337 FLYB: FBst0014337 | FLYB symbol: P{SUPor-P}chico$^{KG00032}$ |
| Genetic reagent (*D. melanogaster*) | Myc-GFP | Bloomington Drosophila Stock Center | BDSC: 81274 FLYB: FBti0147732 | FLYB symbol: PBac{y$^{+}$-attP-3B} VK00033 |
| Genetic reagent (*D. melanogaster*) | Sgg-GFP | Bloomington Drosophila Stock Center | BDSC: 66770 FLYB: FBst0066770 | FLYB symbol: Mi{PT-GFSTF.1} sgg$^{MI11971-GFSTF.1}$ |
| Genetic reagent (*D. melanogaster*) | PDK1-GFP | Bloomington Drosophila Stock Center | BDSC: 59836 FLYB: FBst0059836 | FLYB symbol: Mi{PT-GFSTF.0} Pdk1$^{MI06823-GFSTF.0}$ |
| Genetic reagent (*D. melanogaster*) | myc-Dp110 | Bloomington Drosophila Stock Center | BDSC: 25915 FLYB: FBst0025915 | FLYB symbol: P{Myc-Dp110}1 |
| Genetic reagent (*D. melanogaster*) | Chico-GFP | Vienna Drosophila RNAi Center | VDRC: v318104 FLYB: FBst0491524 | FLYB symbol: PBac{fTRG00456.sfGFP-TVPTBF}VK00033 |
| Genetic reagent (*D. melanogaster*) | InR-EGFP | This paper | HX lab | EGFP inserted before the stop codon of InR genomic locus |
| Genetic reagent (*D. melanogaster*) | UAS-FLP | Bloomington Drosophila Stock Center | BDSC: 29731 FLYB: FBti0128596 | FLYB symbol: P{w[+mC]=UASp FLP.G}3 |
| Genetic reagent (*D. melanogaster*) | TM3, Sb[1], Ser[1]/ TM6B, Tb[1] | Bloomington Drosophila Stock Center | BDSC: 2537 FLYB: FBst0002537 | FLYB symbol: TM3, Sb[1] Ser[1] /TM6B, Tb[1] |
| Antibody | Mouse monoclonal anti-GFP | Roche | 11814460001 | IF(1:1000) |
| Antibody | Mouse monoclonal anti-Hts | Developmental Studies Hybridoma Bank | 1B1 | IF(1:1000) |

*Continued on next page*

*Continued*

| Reagent type (species) or resource | Designation | Source or reference | Identifiers | Additional information |
|---|---|---|---|---|
| Antibody | Mouse monoclonal anti-ATP5A | ATP5A | ab14748 | IF(1:400) |
| Antibody | Rabbit polyclonal anti-GFP | Novus | NB600-308 | IF(1:1000) |
| Antibody | Rabbit polyclonal anti-dMyc | Santa Cruz Biotechnology | d1-717 | IF(1:200) |
| Antibody | Rabbit polyclonal anti-Vasa | Santa Cruz Biotechnology | sc-30210 | IF(1:500) |
| Antibody | Rabbit monoclonal anti-p-GSK3 | Cell Signaling Technology | 9323S | IF(1:200) |
| Antibody | Mouse monoclonal anti-GSK3 | Cell Signaling Technology | 9832S | IF(1:200) |
| Antibody | Rabbit polyclonal anti-AKT | Cell Signaling Technology | 9272S | IF(1:200) |
| Antibody | Rabbit polyclonal anti-p-AKT | Cell Signaling Technology | 4054S | IF(1:200) |
| Antibody | Mouse monoclonal β-galactosidase | Promega | Z378A | IF(1:1000) |
| Peptide, recombinant protein | Cytochrome c from equine heart | Sigma-Aldrich | C7752 | COX histochemistry staining |
| Peptide, recombinant protein | Catalase from bovine liver | Sigma-Aldrich | C9322 | COX histochemistry staining |
| Chemical compound, drug | TMRM | Thermo Fisher Scientific | I34361 | 1:10000 on dissected ovaries |
| Chemical compound, drug | MitoTracker Green | Thermo Fisher Scientific | M7514 | 1:10000 on dissected ovaries |
| Chemical compound, drug | 3,3′-Diaminobenzidine tetrahydrochloride | Sigma-Aldrich | D5905 | COX histochemistry staining |
| Chemical compound, drug | Nitrotetrazolium Blue chloride | Sigma-Aldrich | N6876 | COX/SDH dual histochemistry staining |
| Chemical compound, drug | Phenazine methosulfate | Sigma-Aldrich | P9625 | COX/SDH dual histochemistry staining |
| Chemical compound, drug | Sodium succinate | Sigma-Aldrich | S2378 | COX/SDH dual histochemistry staining |
| Chemical compound, drug | Heparin ammonium salt | Sigma-Aldrich | H6279 | RNA FISH |
| Other | Salmon Sperm DNA | Invitrogen | 15632011 | RNA FISH |
| Commercial assay, kit | Click-iT Plus EdU Alexa Fluor 488 Imaging Kit | Thermo Fisher Scientific | C10637 | labeling replicating mtDNA |
| Commercial assay, kit | Ribo-Zero rRNA Removal Kit | Illumina Inc. | MRZH116 | Used in RNA preparation for RNAseq |
| Sequence-based reagent | *InR* chiRNA targeting sequence | This paper | HX lab | CCTTTCCGTAGAT GGATGACACC |
| Sequence-based reagent | InR-F1 | This paper | HX lab | ATGATGTCATCGGT GGGTCCTCAC |

*Continued on next page*

*Continued*

| Reagent type (species) or resource | Designation | Source or reference | Identifiers | Additional information |
|---|---|---|---|---|
| Sequence-based reagent | EGFP-seqR | This paper | HX lab | CTTGTAGTTGCC GTCGTCCTTGAA |
| Sequence-based reagent | InR-F2 | This paper | HX lab | AGCACATTGTG TCAGTCTTCG |
| Sequence-based reagent | InR-R | This paper | HX lab | CTCATTTTCCGAA GCTTGGCTTCC |
| Sequence-based reagent | *mt:CoI*, Xho1 site genotyping F | *Hill et al., 2014* | https://doi.org/ 10.1038/ng.2920 | TGGAGCTATTG GAGGACTAAATCA |
| Sequence-based reagent | *mt:CoI*, Xho1 site genotyping R | *Hill et al., 2014* | https://doi.org/ 10.1038/ng.2920 | GCTCCTGTTAA TGGTCATGGACT |
| Sequence-based reagent | qPCR, *his4*-F | *Zhang et al., 2015* | https://doi.org/10.15252/ embr.201439636 | TCCAAGGTATCACGAAGCC |
| Sequence-based reagent | qPCR, *his4*-R | (*Zhang et al., 2015*) | https://doi.org/10.15252/ embr.201439636 | AACCTTCAGAACGCCAC |
| Sequence-based reagent | qPCR, *mt:CoI*-F | (*Zhang et al., 2015*) | https://doi.org/10.15252/ embr.201439636 | ATTGGAGTTAATTT AACATTTTTTCCTCA |
| Sequence-based reagent | qPCR, *mt:CoI*-R | *Zhang et al., 2015* | https://doi.org/10.15252/ embr.201439636 | AGTTGATACAATATTT CATGTTGTGTAAG |
| Software, algorithm | ImageJ | NIH | https://imagej.nih.gov/ij/ | |
| Scoftware, algorithm | GraphPad Prism7 | GraphPad Software, Inc | http://www. graphpad.com/ | |

## Fly stocks and genetics

Flies were maintained on standard BDSC cornmeal medium at 25°C. RNAi lines for candidate screen are listed in *Supplementary file 1*. Embryo hatch assay was performed as previously described (*Zhang et al., 2019*). To control the genetic background, chico[KG], chico[1], ts and myc[P0] strains were back-crossed with w[1118] for six generations.

## CRISPR/Cas9 in flies

To tag *InR* with EGFP at its endogenous locus, a targeting cassette comprising of 1 kb upstream of InR stop codon, *EGFP* ORF, a fragment containing *GMR-Hid* flanked by two FRT sites, and 1 kb downstream of *InR* stop codon was inserted into a pOT2 vector. This donor construct and a *InR* chiRNA construct were injected into the embryos of M{vas-Cas9}ZH-2A (BL51323) by Bestgene Inc.. G0 adults were crossed with w1118, and progeny with small eye phenotype were selected as candidates due to the expression of *GMR-Hid*. Insertion events were further confirmed by PCR using two pairs of primers: 1, InR-F1/EGFP-seqR and 2, InR-F2/InR-R. To remove the *GMR-Hid* cassette, the *InR-EGFP-GMR-Hid* flies were crossed with *nos-Gal4; UASp-FLP*. The F1 progeny with the genotype of *nos-Gal4/+; UASp-FLP/InR-EGFP-GMR-Hid* were selected and crossed with *TM3, Sb1, Ser1/ TM6B, Tb1*. The F2 flies of *InR-EGFP/TM6B, Tb1* with normal white eyes were selected and maintained.

## Measurement of mtDNA copy number and quantification of heteroplasmy

Total DNA was isolated from eggs with QIAamp DNA Micro Kit (Qiagen). The mtDNA copy number was measured by quantitative real-time PCR with primers targeting to cytochrome c oxidase subunit I (mt:CoI) and His4 genes. Quantification of heteroplasmy was performed as described previously by qPCR (*Zhang et al., 2019*), with primer pairs: his4-F/his4-R: and mt:CoI-F/mt:CoI-R. Heteroplasmic female flies were transferred from 18°C to 29°C after eclosion. Each female was mated with five wt males. Ten eggs produced from the day 7 at 29°C were collected. The genomic DNA from female flies and eggs was extracted and their heteroplasmy levels were determined as shown before (*Hill et al., 2014*), with primers: *mt:CoI*, Xho1 site genotyping F and *mt:CoI*, Xho1 site genotyping R.

## RNA sequencing and RNA-seq analysis

Total RNA was extracted by Trizol (Life Technologies) following its standard protocol. Poly (A) capture libraries were generated at the DNA Sequencing and Genomics Core, NHLBI, NIH. RNA sequencing was performed with using an Hiseq3000 (Illumina) and 75 bp pair-end reads were generated at the DNA Sequencing and Genomics Core, NHLBI, NIH. Raw sequence reads were quality-trimmed using Trim Galore! (v0.3.7) and aligned using HISAT2 against the Dm6 reference genome. Uniquely mapped paired-end reads were then used for subsequent analyses. FeatureCounts was used for gene level abundance estimation. Principal component analysis (PCA) was used to assess outlier samples. Genes were kept in the analysis if they had Counts Per Million (CPM > 1) in at least half the samples. We adjusted for multiple testing by reporting the FDR q-values for each feature. Features with q < 5% were declared as genome-wide significant. Genes with three or more-fold changes on mRNA level in $myc^{P0}$ mutant compared with wt were considered differentially expressed. Gene Ontology (GO) was used to analyze gene set enrichment. FDR q-values were estimated to correct the p-values for the multiple testing issue.

To evaluate the enrichment of genes associated with mitochondrial processes in our decreased genes, we performed an approximate permutation test or resampling approach (*Churchill and Doerge, 1994*; *Good, 2005*; *Peter and Westfall, 1993*), with null hypothesis that there is no difference between expression of mitochondrial and non-mitochondrial genes. The approach was performed by randomly selecting an equal number of genes (4,733) from the entire observed genome that were expressed in our study and were matched by gene size (±10 kb) as well as the GC content (±20%) with our observed genes. We repeated this process 10,000 times to calculate the empirical p-value, the number of times that the number of mitochondrial genes in the randomly drawn genes was greater than what was found in the observed set (458). The permutation empirical p-value was $p<10^{-4}$ indicating that there is a significant enrichment for genes associated with mitochondrial processes in our decreased gene list.

Myc target analysis dMyc enriched binding regions that is peak files were downloaded from the Gene Expression Omnibus (GEO) repository (*Barrett et al., 2013*; *Edgar et al., 2002*). The accession numbers of the downloaded files are GSE53560, GSE53559, and GSE49774 (*Slattery et al., 2014*). Using an in-house R script and Bedtools utility (*Quinlan and Hall, 2010*), the peaks were annotated to *Drosophila melanogaster*'s genome using Ensembl V(91) annotation (*Aken et al., 2017*) to identify downstream targets of dMyc. If a ChIP-seq peak from any one of the datasets was observed between 250 base-pairs upstream to 100 base-pair downstream of a gene's transcription start site, then the gene was assumed to be putative downstream targets of dMyc. To reduce the false positives that could potentially arise due to different origins of tissues in RNA-seq and ChIP-seq data, only those genes that were both differentially expressed and had a ChIP-seq peak were considered as putative targets of dMYC.

## Live image

Live image of fly ovaries was performed as previously reported (*Zhang et al., 2019*). Ovaries from wt flies were stained with TMRM (1:10000, I34361, Thermo Fisher Scientific) and MTgreen (1:10000, M7514, Thermo Fisher Scientific) in PBS for 20 min followed by PBS washes for three times. Ovaries from Sgg-GFP flies were dissected. Ovarioles were isolated and immerged in halocarbon oil on coverslips, then live imaged with a Perkin Elmer Ultraview system. The ratiometric image was generated with ImageJ, which the intensity of red channel is divided by that of green channel.

## SDH and COX activity staining in the ovary

To determine if changes in the intensities of COX staining linearly or exponentially correlate with the activities/amounts of complexes IV in the ovary, we generated standards for COX staining by blotting various amount of Cytochrome c oxidase (C5499, Sigma, for the COX standards) on a nitrocellulose membrane with a slot blotting apparatus (Bio-Dot SF Apparatus). COX staining solution contains 50 mM phosphate (pH 7.4), 4 mM 3,3'-diaminobenzidine, 2 µg/ml catalase, 200 µM cytochrome c, 84 mM malonate, 60 µM rotenone and 4 mM antimycin A. The nitrocellulose membrane blotted with COX was incubated in the COX staining solution at room temperature for 30 min. The reaction was stopped by incubation with 4% paraformaldehyde for 15 mins. Membrane was then washed twice in 50 mM phosphate (pH 7.4) and scanned. The intensities of bands were quantified

using the 'Gels' function in ImageJ. Band intensities and the according amounts of mitochondria or Cytochrome c oxidase were plotted in Excel. Trendlines were generated based on linear or exponential relation, respectively. We found that $R^2$ values of linear correlation are 0.9919 for COX activity standards, while $R^2$ values of exponential correlation are reduced to 0.8658.

For COX/SDH dual activity staining, five pairs of ovaries were dissected in PBS and intact ovarioles were separated with a dissection needle. Ovaries were incubated alive with COX staining solution for 30 min, followed by three washes in 50 mM phosphate (pH 7.4) for 5 min each. Ovaries were then incubated for 10 min in the SDH staining solution, containing 50 mM phosphate (pH 7.4), 42 mM succinic acid, 0.4 mM phenazine methosulfate, 0.5 mM nitro blue tetrazolium, 4.5 mM EDTA, 60 µM rotenone, 4 mM antimycin A and 2 mM KCN. Ovaries were then washed twice in 50 mM phosphate (pH 7.4) for 5 min each and 4% paraformaldehyde fixation for 15 mins. Then, ovaries were washed for twice in 50 mM phosphate (pH 7.4) for 5 min each and immersed in 80% glycerol in 50 mM phosphate (pH 7.4). For COX single activity staining, 8 to 10 pairs of ovaries were dissected in PBS and intact ovarioles were separated with a dissection needle. Ovaries from the same genotype were divided into two groups and incubated alive with COX staining solution for 30 min. No saturation was observed for either staining. For the negative control, ovaries ($w^{1118}$ or $luciferase^{RNAi}$) were incubated with COX staining solution added with a complex IV inhibitor (2 mM KCN). The negative controls of COX staining were performed for each batch of ETC activity staining. The reactions were followed by two washes in 50 mM phosphate (pH 7.4) for 5 min each and 4% paraformaldehyde fixation for 15 min. Then, ovaries were washed for twice in 50 mM phosphate (pH 7.4) for 5 min each and immersed in 80% glycerol in 50 mM phosphate (pH 7.4). Brightfield germarium images were collected by Zeiss Axio Observer Z1 microscope.

Quantification for relative ETC activities in the germarium was performed using ImageJ. From the opened images, germline cysts were isolated by 'Clear Outside'. Color of the cysts was inverted and converted into gray. 'Color Threshold' was used to select COX staining. Intensity of selected staining area was measured. Intensity of non-selected area was considered as background and subtracted from the intensity of selected area. From each batch of activity staining, the COX activity was calculated by normalizing the staining intensities in the germarium to the COX activity standard curves. The intensity from the negative control group, with addition of inhibitor for COX, was considered as '0' activity, while the intensity from the control groups was considered as a relative '1' activity.

## Immunofluorescence staining and fluorescence in situ hybridization (FISH)

Ovary dissection, immunostaining, and EdU incorporation were performed as described before (*Hill et al., 2014*). Click-iT Plus EdU Alexa Fluor 488 Imaging Kit (C10637, Thermo Fisher Scientific) was used for EdU incorporation and visualization. Stellaris FISH probes against, *Cyt-C1*, *Cyt-B*, *CoxIV*, *CoxIII*, or *InR* mRNA were synthesized from Biosearch Technologies. Sequences of the probes are listed in the *Supplementary file 7*. FISH of *Drosophila* ovaries was conducted as previously described (*Trcek et al., 2017*). Confocal images were collected by a Perkin Elmer Ultraview system or Instant Sim (iSIM) Super-Resolution Microscope. All images were processed with Image J.

## Quantification and statistical analysis

All statistical analyses were conducted with Prism 7 (GraphPad Software). Error bars in all charts represent standard errors. p-Values were performed with Two-tailed Student's t test. Statistical significance of difference was considered when $p < 0.05$.

## Data availability

The data were deposited in Gene Expression Omnibus of NCBI (*Edgar et al., 2002*) and will be available with accession number (GEO: GSE126997).

## Acknowledgements

We thank F Chanut for comments and editing on the manuscript; E Geisbrecht, L Johnston, E Moreno, O Schuldiner, the Bloomington *Drosophila* Stock Center and the Vienna Drosophila Resource Center for various fly lines; the Developmental Studies of Hybridoma Bank for various antibodies;

the NHLBI DNA Sequencing and Genomics Core and Bestgene Inc for technical assistance. This work is supported by NHLBI Intramural Research Program.

## Additional information

### Funding

| Funder | Grant reference number | Author |
|---|---|---|
| National Heart, Lung, and Blood Institute | 1ZIAHL006153-07 | Hong Xu |

The funders had no role in study design, data collection and interpretation, or the decision to submit the work for publication.

### Author contributions

Zong-Heng Wang, Conceptualization, Data curation, Software, Formal analysis, Validation, Investigation, Methodology, Writing—original draft, Project administration, Writing—review and editing; Yi Liu, Data curation; Vijender Chaitankar, Software, Formal analysis, Validation, Methodology, Writing—review and editing; Mehdi Pirooznia, Software, Formal analysis, Supervision, Validation, Methodology, Writing—review and editing; Hong Xu, Conceptualization, Resources, Supervision, Funding acquisition, Validation, Methodology, Writing—original draft, Project administration, Writing—review and editing

### Author ORCIDs

Hong Xu (ID) https://orcid.org/0000-0002-1423-1809

### Decision letter and Author response

Decision letter https://doi.org/10.7554/eLife.49309.042
Author response https://doi.org/10.7554/eLife.49309.043

## Additional files

### Supplementary files

• Supplementary file 1. List of RNAi lines and corresponding genes in the candidate RNAi screen.
DOI: https://doi.org/10.7554/eLife.49309.025

• Supplementary file 2. List of Differentially expressed genes between $myc^{P0}$ and wt ovaries.
DOI: https://doi.org/10.7554/eLife.49309.026

• Supplementary file 3. Gene ontology analysis downregulated genes in $myc^{P0}$ ovaries.
DOI: https://doi.org/10.7554/eLife.49309.027

• Supplementary file 4. Gene ontology analysis downregulated mitochondrial genes in $myc^{P0}$ ovaries.
DOI: https://doi.org/10.7554/eLife.49309.028

• Supplementary file 5. List of transcription factors that are downregulated in $myc^{P0}$ ovaries.
DOI: https://doi.org/10.7554/eLife.49309.029

• Supplementary file 6. List of conserved Myc targets in human and flies.
DOI: https://doi.org/10.7554/eLife.49309.030

• Supplementary file 7. List of fluorescence DNA probes for FISH.
DOI: https://doi.org/10.7554/eLife.49309.031

• Transparent reporting form   DOI: https://doi.org/10.7554/eLife.49309.032

### Data availability

The data were deposited in Gene Expression Omnibus of NCBI and will be available with accession number (GEO: GSE126997).

The following dataset was generated:

**Database and**

| Author(s) | Year | Dataset title | Dataset URL | Identifier |
|---|---|---|---|---|
| Wang Z-H, Liu Y, Chaitankar V, Pirooznia M, Xu H | 2019 | Myc regulation of ETC Biogenesis and mtDNA Replication | https://www.ncbi.nlm.nih.gov/geo/query/acc.cgi?acc=GSE126997 | NCBI Gene Expression Omnibus, GSE126997 |

The following previously published datasets were used:

| Author(s) | Year | Dataset title | Dataset URL | Database and Identifier |
|---|---|---|---|---|
| White K, Ma L, Slattery M | 2014 | dMyc_S2_cells_ChIP-seq | https://www.ncbi.nlm.nih.gov/geo/query/acc.cgi?acc=GSE53560 | NCBI Gene Expression Omnibus, GSE53560 |
| White K, Ma L, Slattery M | 2014 | dMyc_Kc167_cells_ChIP-seq | https://www.ncbi.nlm.nih.gov/geo/query/acc.cgi?acc=GSE53559 | NCBI Gene Expression Omnibus, GSE53559 |
| White K, Ma L, Slattery M | 2014 | dMyc_W3L_ChIP-seq | https://www.ncbi.nlm.nih.gov/geo/query/acc.cgi?acc=GSE49774 | NCBI Gene Expression Omnibus, GSE49774 |

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
