## [Decision Letter]

Thank you for submitting your article "Electron transport chain biogenesis activated by a JNK-insulin-Myc relay primes mitochondrial inheritance in *Drosophila*" for consideration by *eLife*. Your article has been reviewed by three peer reviewers, and the evaluation has been overseen by Hugo Bellen as the Reviewing Editor and Michael Eisen as the Senior Editor. The reviewers have opted to remain anonymous.

The reviewers have discussed the reviews with one another and the Reviewing Editor has drafted this decision to help you prepare a revised submission.

Summary:

In this manuscript, Wang et al. further describe the role of insulin/GSK3 signaling in the regulation of mitochondria during germ cell development. While insulin/GSK3 signaling has been previously shown to have a major effect on germ line mitochondrial activity, this study highlights the extensive interplay between the pathway and mitochondrial function by showing the pathway is required for mtDNA replication. The authors extend these findings to show that insulin-Myc and JNK signaling coordinate germ cell mitochondrial biogenesis with developmental progression. Overall this study addresses a fundamental question in germ cell biology and provides valuable insight into how developmental signaling pathways coordinate mitochondrial biogenesis with developmental progression. However, the authors overstate their results regarding the specific effects on ETC regulation and mitochondrial inheritance. The authors also lack a few controls and rescue experiments that support their key findings. Once these issues are corrected this study will provide a valuable contribution to our understanding of how mitochondrial biogenesis is coupled to early germ cell development.

Positive aspects of the paper:

• The authors show clear and significant roles of insulin, GSK3, Myc, and JNK signaling on mitochondrial replication in early developing germ cells.

• The phenotypes described are highly consistent with previous studies that describe the molecular mechanisms that underlie the insulin/GSK3/Myc pathway.

• The authors also nicely show that the changes in mtDNA they describe correspond with reduced levels of SDH and COX activity.

• The authors show, consistent with previous studies from Chi V. Dang's lab, that Myc plays an in vivo role in the regulation of mitochondrial gene expression and mitochondrial biogenesis.

Overall weaknesses:

• The authors imply that JNK-insulin-Myc signaling has specific roles in ETC activity, mtDNA replication, and mitochondrial inheritance. However, the most consistent model is that the effects on ETC activity and mitochondrial inheritance the authors observed simply stem from changes in mitochondrial number.

• The authors don't provide adequate genetic background controls for many of their key experiments.

• The authors fail to adequately present what is known from previous literature regarding Myc regulation of mitochondrial biogenesis and GSK3 regulation of Myc.

• The manuscripts title implies the story is about ETC biogenesis and mitochondrial inheritance however ETC biogenesis is not directly examined and mitochondrial inheritance is only examined directly in one panel in Figure 7. The title of the manuscript should be changed to reflect the major contribution of this paper in understanding how insulin-Myc controls mitochondrial biogenesis.

Essential revisions:

- The major focus of this study is the regulation of mitochondrial biogenesis and metabolism by the transcription factor Myc. The authors identify Myc as a regulator of SDH/COX activity based on a targeted RNAi screen. Subsequent RNAseq analysis revealed that Myc transcriptionally regulates nearly all of mitochondrial metabolism. However, a quick analysis of the RNAseq data in Supplementary File 2 reveals that 1/3 of the genome exhibit a more than 3-fold decrease in expression. The text then states that there is a significant enrichment for genes associated with mitochondrial processes, however, there is no statistical analysis supporting this claim nor is there a list of mitochondrial genes used for the analysis. Both pieces of data are required to support such a conclusion.

- The authors use "wt" flies as controls for many of their experiments. Given mitochondrial function and cellular metabolism can be heavily influenced by genetic background the authors should either rescue the key phenotypes (COXIV activity, SDH activity, and mitochondrial EdU incorporation) associated with *chico* and Myc they describe in the text, or provide more suitable genetically matched controls for their key experiments.

-The authors provide a rudimentary description of the gene expression data presented in Figure 2. Given the authors belief that Myc is the key factor downstream of insulin signaling driving mitochondrial biogenesis, this data set should be analyzed in much more depth. It would also be helpful to examine how many of these Myc-regulated genes are in the mammalian data sets such as Li et al. (2005).

- The study has major deficiencies related to the COX/SDH staining assays:

1) To my knowledge, this is the first example of this classic assay being used in *Drosophila*. While I applaud these efforts, the authors must provide a detailed description of how they conducted the assays. For example, how long were tissues fixed? How long were tissues stained? These details are essential for evaluating the method.

2) As described in the methods paper cited by the authors (Ross, 2011), this type of analysis requires specificity controls for both COX activity and SDH activity. Please provide this data. Note that these controls are particularly important in light of Supplementary Figure 2A, where the germarium still exhibits staining despite the use of *CoVa* RNAi.

3) Nearly every figure contains a panel labeled "relative ETC activity," yet there is no description of how this is value was calculated in the text. My best guess is that this measurement is based on the intensity of COX/SDH staining. If so, this data poses a number of problems. To conduct this type of analysis, each experiment must include a standard curve of COX activity. Moreover, each assay must also include negative controls. See (Melendez-Ferro et al., 2013) of the care that must be taken when making comparisons of COX activity across tissues samples. Based on the text included within the manuscript, I see no evidence of these controls. If the authors indeed failed to conduct these controls, this is a major issue that undermines all of the "ETC activity" panels.

4) Also in regards to "relative ETC activity," I'm not sure what this term means. Do the authors mean relative combined SDH/COX activity?

5) I'm also concerned if total staining intensity was used as a readout. The combined COX/SDH assay produces to different colored compounds. Blue is indicative of decreased COX activity.

If the "ETC activity" measurements really represent intensity of COX/SDH staining, then the authors must replace these panels with some other evidence of changes in electron transport chain activity.

- The authors make statements that imply insulin-Myc, JNK signaling have specific roles in regulating mitochondrial biogenesis, ETC biogenesis, and inheritance. However, the authors' data suggest the observed effects on mitochondrial activity may simply reflect changes in mitochondrial biogenesis. This is a bit misleading because it implies the pathway may have a more sophisticated role in mitochondrial function than the data supports. The authors should either revise the manuscript to reflect this more simplified view or provide more clear-cut data that shows specific changes in ETC activity on a per mitochondria basis.

Similarly, the observed changes in mitochondrial inheritance could be due to defects in amplifying healthy mitochondria. Given total mitochondrial number is decreased so dramatically in *chico* and *bsk* mutant eggs, it is difficult to make strong conclusions about mitochondrial inheritance given the data presented. As a result, the authors should either revise the text to provide a more simple discussion of the data or provide evidence that examines whether the transmission of defective mitochondria in a heteroplasmic fly is under direct control of insulin-Myc signaling or simply a result of defective mitochondrial biogenesis in germ cells.

---

## [Author Response]

Essential revisions:- The major focus of this study is the regulation of mitochondrial biogenesis and metabolism by the transcription factor Myc. The authors identify Myc as a regulator of SDH/COX activity based on a targeted RNAi screen. Subsequent RNAseq analysis revealed that Myc transcriptionally regulates nearly all of mitochondrial metabolism. However, a quick analysis of the RNAseq data in Supplementary File 2 reveals that 1/3 of the genome exhibit a more than 3-fold decrease in expression. The text then states that there is a significant enrichment for genes associated with mitochondrial processes, however, there is no statistical analysis supporting this claim nor is there a list of mitochondrial genes used for the analysis. Both pieces of data are required to support such a conclusion.

We previously published the list of nuclear-encoded mitochondrial genes in the *Drosophila* genome (Zhang et al., 2019). We used this list as the reference to evaluate the significant enrichment of genes associated with mitochondrial processes in the context of genes affected in *myc* mutant ovaries, and carried out a gene ontology analysis for the enrichment of cellular pathways. The detailed procedure of the bioinformatic analysis has been included in the revised Materials and methods section. We now include the list of genes and the statistical analyses in the Supplementary Files 3-5, and describe the procedure and result of the cellular pathway enrichment analysis in the Materials and methods.

- The authors use "wt" flies as controls for many of their experiments. Given mitochondrial function and cellular metabolism can be heavily influenced by genetic background the authors should either rescue the key phenotypes (COXIV activity, SDH activity, and mitochondrial EdU incorporation) associated with chico and Myc they describe in the text, or provide more suitable genetically matched controls for their key experiments.

The *chico* and *myc* mutants have been backcrossed to *w^1118^*to clean up the genetic background, and *w^1118^*flies were used as the control in these experiments. We clarify this issue in the revised Materials and methods section.

-The authors provide a rudimentary description of the gene expression data presented in Figure 2. Given the authors belief that Myc is the key factor downstream of insulin signaling driving mitochondrial biogenesis, this data set should be analyzed in much more depth. It would also be helpful to examine how many of these Myc-regulated genes are in the mammalian data sets such as Li et al. (2005).

As we addressed a previous comment on the RNAseq data analysis, we have re-analyzed RNAseq data and include the detailed analyses in the Supplementary File 3.

In the previous study (Li et al., 2005), a total of 2679 genes were reported to be regulated by Myc in a human cell line. However, only 198 mitochondrial genes emerged from this study, presumably due to the lack of a full compendium of mitochondrial genes (MitoCarta) back then. Also, the full list of Myc-regulated genes is not available from that study (Li et al., 2005), which prevents us from carrying out further analyses with the mammalian data set. Nonetheless, we looked into the 198 Myc-regulated mitochondrial genes, and found that 185 of them have homologs in *Drosophila*. Importantly, 44.9% (101 out of 225) of the fly homologs are down-regulated in *myc^P0^*mutant ovaries. This analysis indeed demonstrates an evolutionarily conserved function of Myc in regulating mitochondrial biogenesis. We discuss this comparative analysis in the second paragraph of the Discussion, and include the dataset in the Supplementary File 6.

- The study has major deficiencies related to the COX/SDH staining assays:1) To my knowledge, this is the first example of this classic assay being used in Drosophila. While I applaud these efforts, the authors must provide a detailed description of how they conducted the assays. For example, how long were tissues fixed? How long were tissues stained? These details are essential for evaluating the method.

We now describe the experimental procedure in detail in the Materials and methods subsection “SDH and COX activity staining in the ovary”.

2) As described in the methods paper cited by the authors (Ross, 2011), this type of analysis requires specificity controls for both COX activity and SDH activity. Please provide this data. Note that these controls are particularly important in light of Supplementary Figure 2A, where the germarium still exhibits staining despite the use of CoVa RNAi.

Complex IV (COX, brown staining) is disrupted by *CoVa* RNAi, while complex II remains intact. Thus, the staining in the germarium of *CoVa* RNAi is predominantly derived from the SDH activity (blue color). We have moved these data to the revised Figure 1—figure supplement 1C.

3) Nearly every figure contains a panel labeled "relative ETC activity," yet there is no description of how this is value was calculated in the text. My best guess is that this measurement is based on the intensity of COX/SDH staining. If so, this data poses a number of problems. To conduct this type of analysis, each experiment must include a standard curve of COX activity. Moreover, each assay must also include negative controls. See (Melendez-Ferro et al., 2013) of the care that must be taken when making comparisons of COX activity across tissues samples. Based on the text included within the manuscript, I see no evidence of these controls. If the authors indeed failed to conduct these controls, this is a major issue that undermines all of the "ETC activity" panels.4) Also in regards to "relative ETC activity," I'm not sure what this term means. Do the authors mean relative combined SDH/COX activity?5) I'm also concerned if total staining intensity was used as a readout. The combined COX/SDH assay produces to different colored compounds. Blue is indicative of decreased COX activity.If the "ETC activity" measurements really represent intensity of COX/SDH staining, then the authors must replace these panels with some other evidence of changes in electron transport chain activity.

We agree with the reviewers that the combined COX/SDH assay is best suited for quantification, even though we noticed that the majority of color deposit was derived from COX activity stanning in the COX/SDH dual staining assay. In the revision, we carried out COX staining along, and used the COX activity as an indication of ETC activity.

We applied the KCN, the inhibitor of COX in the experiments to confirm the specificity of COX activity staining. We also generated the standard curve of COX, adopting the procedure used in the paper suggested by the reviewer. The optical density of COX activity staining fits a linear relationship with the amount of cytochrome C oxidase (Sigma) used in the assay. We therefore used OD as a means to semi-quantify the relative SDH or COX activity. We revised the figure to include the new data, and describe the experimental procedure in the Materials and methods section.

- The authors make statements that imply insulin-Myc, JNK signaling have specific roles in regulating mitochondrial biogenesis, ETC biogenesis, and inheritance. However, the authors’ data suggest the observed effects on mitochondrial activity may simply reflect changes in mitochondrial biogenesis. This is a bit misleading because it implies the pathway may have a more sophisticated role in mitochondrial function than the data supports. The authors should either revise the manuscript to reflect this more simplified view or provide more clear-cut data that shows specific changes in ETC activity on a per mitochondria basis.Similarly, the observed changes in mitochondrial inheritance could be due to defects in amplifying healthy mitochondria. Given total mitochondrial number is decreased so dramatically in chico and bsk mutant eggs, it is difficult to make strong conclusions about mitochondrial inheritance given the data presented. As a result, the authors should either revise the text to provide a more simple discussion of the data or provide evidence that examines whether the transmission of defective mitochondria in a heteroplasmic fly is under direct control of insulin-myc signaling or simply a result of defective mitochondrial biogenesis in germ cells.

Previous studies demonstrate that the prodigious mitochondrial biogenesis during oogenesis not only furnishes mature oocytes with adequate amount of mitochondria, but also allows proliferation competition, which limits the transmission of deleterious mtDNA mutations (Zhang et al., 2019).

We revised the manuscript to reflect that insulin-Myc signaling primarily promotes mitochondrial biogenesis, through which it can influence mitochondrial inheritance both quantitively and quantitively. It now reads “This signaling relay promotes mitochondrial biogenesis in the ovary, and thereby plays a role in limiting the transmission of deleterious mtDNA mutations.”.